# Complement opsonization of HIV affects primary infection of human colorectal mucosa and subsequent activation of T cells

Pradyot Bhattacharya[1], Rada Ellegård[1‡], Mohammad Khalid[1], Cecilia Svanberg[1], Melissa Govender[1], Åsa V Keita[2], Johan D Söderholm[2], Pär Myrelid[2], Esaki M Shankar[3,4†], Sofia Nyström[1,5†], Marie Larsson[1*]

[1]Division of Molecular Medicine and Virology, Department of Clinical and Experimental Medicine, Linköping University, Linköping, Sweden; [2]Division of Surgery, Orthopedics and Oncology, Linköping University, Linköping, Sweden; [3]Center of Excellence for Research in AIDS (CERiA), University of Malaya, Lembah Pantai, Kuala Lumpur, Malaysia; [4]Division of Infection Biology and Medical Microbiology, Department of Life Sciences, Central University of Tamil Nadu, Thiruvarur, India; [5]Department of Clinical Immunology and Transfusion Medicine and Department of Clinical and Experimental Medicine, Linköping University, Linköping, Sweden

*For correspondence:
marie.larsson@liu.se

†These authors contributed equally to this work

Present address: ‡Division of Clinical Genetics, and Department of Biomedical and Clinical Sciences, Linköping University, Linköping, Sweden

Competing interests: The authors declare that no competing interests exist.

**Abstract** HIV transmission via genital and colorectal mucosa are the most common routes of dissemination. Here, we explored the effects of free and complement-opsonized HIV on colorectal tissue. Initially, there was higher antiviral responses in the free HIV compared to complement-opsonized virus. The mucosal transcriptional response at 24 hr revealed the involvement of activated T cells, which was mirrored in cellular responses observed at 96 hr in isolated mucosal T cells. Further, HIV exposure led to skewing of T cell phenotypes predominantly to inflammatory CD4+ T cells, that is Th17 and Th1Th17 subsets. Of note, HIV exposure created an environment that altered the CD8+ T cell phenotype, for example expression of regulatory factors, especially when the virions were opsonized with complement factors. Our findings suggest that HIV-opsonization alters the activation and signaling pathways in the colorectal mucosa, which promotes viral establishment by creating an environment that stimulates mucosal T cell activation and inflammatory Th cells.

## Introduction

Human immunodeficiency virus type 1 (HIV-1) infection, despite the relatively low rates of transmission, remains one of the major global public health challenges (*Stax et al., 2015*). Recent global estimates suggest that in 2017 alone,~1.8 million individuals became infected with HIV giving rise to a total of 36.9 (31.1–43.9) million HIV-infected individuals (*UNAIDS, 2017*). Interestingly, despite an increase in the rates of accessibility to antiretroviral therapy (ART), progression to acquired immunodeficiency syndrome (AIDS) resulted in ~0.95 (0.67–1.3) million deaths worldwide in 2017 (*UNAIDS, 2017*). Sexual intercourse is the primary route of viral entry, and the risk of transmission via rectal mucosa is reportedly ~10 times higher among individuals who practiced receptive anal intercourse than those engaged in vaginocervical intercourse (*Hladik and McElrath, 2008*). The enhanced susceptibility of colorectal mucosa to HIV infection appears to be attributed to the

structure and fragility of the epithelial layer as well as the activated nature of lymphocytes present in the colon (*Elliott et al., 2018*; *McElrath et al., 2013*; *Grivel et al., 2010*).

Following its introduction via semen, the virus breaches the rectal mucosal barrier either through ruptures caused during intercourse (*Stax et al., 2015*; *Grivel et al., 2010*), or via uptake by dendritic cells (DCs) (*Grivel et al., 2010*). The presence of pre-existing inflammatory conditions such as proctitis (*Bissessor et al., 2013*) or infection by other pathogens such as Herpes simplex virus type 2 (HSV-2) and human papillomavirus (HPV) (*Freeman et al., 2006*; *Welling et al., 2015*) could enhance the rates of mucosal susceptibility to HIV infection. DCs play a key role in dissemination by mediating rapid HIV transfer to CD4+ T cells in the draining lymph nodes (*McDonald, 2010*; *Preza et al., 2014*; *Saba et al., 2010*). Host cell binding and incorporation of HIV is aided by different C-type lectin receptors, such as DC-SIGN, DCIR and MR expressed by DCs. Subsequently, infection of CD4+ T cells takes place through interaction between viral gp120 and T cell co-receptors such as CCR5 or CXCR4. Apart from transferring virions to bystander cells, DCs could also become infected with HIV although the level of infection varies with the DC subtype and maturation status (*Bajtay et al., 2004*; *Izquierdo-Useros et al., 2010*). Other antigen presenting cells like tissue macrophages in colorectal mucosa are also important from the disease point-of-view by functioning as key viral reservoirs (*Brown and Mattapallil, 2014*), not merely disseminating virus to T cells over a longer duration but also to DCs across mucosal tissues (*Brown and Mattapallil, 2014*). HIV, owing to its potential to activate several surface and cytosolic pattern recognition receptors, triggers a local response in immune cells such as induction of IFN-γ, IL-12, TNF, IL-1β (*Sabado et al., 2010*; *McGowan et al., 2004*) and the inflammatory levels appear to correlate with viral replication in the gastrointestinal (GI) tract (*McGowan et al., 2004*; *Brenchley and Douek, 2008*). Interestingly, the severe depletion of intestinal CD4+ T cells that occur in the GI tract during HIV infection shifts their activation status from naïve to effector phenotypes (*Tanko et al., 2018*). Subsequently, the disruption in cellular homeostasis observed during the early stages of infection is likely to contribute to immune activation (*Mehandru et al., 2004*; *Brenchley et al., 2004*).

Th17 cells play a paramount role in maintaining mucosal immune responses against foreign invaders via secretion of inflammatory mediators, antimicrobial peptides and neutrophil recruitment (*O'Connor et al., 2010*). DCs exposed to complement opsonized HIV has been shown to activate Th17 polarization in naïve T cells (*Wilflingseder et al., 2015*). Characterized by the transcriptional expression of RORγt and secretion of IL-17A, IL-17F, and IL-22, Th17 cells are preferentially depleted during chronic HIV infection, a phenomenon directly linked to T-cell activation and HIV-DNA levels in the intestine (*Klatt et al., 2013a*). More recently, another CD4+ T-cell subset Th1Th17, positive for CXCR3 and CCR6, as well as expressing higher levels of both T cell-associated transcription factor (Tbet) and RAR-related orphan receptor-γt (RORγt), has received attention owing to its likely role in HIV pathogenesis by serving as stable viral reservoirs despite the initiation of ART (*Sun et al., 2015*). Together with its deteriorating effect on CD4+ T-cell subsets, HIV infection can also promote memory CD8+ T cell differentiation thereby augmenting the expansion of effector populations (*Tanko et al., 2018*). In addition, the levels of activated CD8+ T cells expressing CD38 have been reported to positively correlate with the amount of HIV RNA in the colon (*Cassol et al., 2010*; *Crowell et al., 2016*). Both CD4+ as well as CD8+ T cells are under the regulation of T-box transcription factors, Tbet and Eomesodermin (EOMES) for defining their effector/memory functions (*Li et al., 2013*). It has been reported that high frequencies of effector CD8+ T cells express Tbet and are often positive for EOMES (*Knox et al., 2014*). To a certain extent, the situation is also similar to that reported in CD4+ T cells, although only a part of Tbet+ CD4+ effector T cells are Eomes co-positive (*Knox et al., 2014*). On the other hand, the functional nature of these CD4+ or CD8+ T-cell subsets upon infection with HIV could be predicted from the expression of several negative immune checkpoint molecules, for instance PD-1, TIM3, LAG3, and CD160 (*Hoffmann et al., 2016*; *Trautmann et al., 2006*), which all correlate with HIV viral load in the host (*Day et al., 2006*). Furthermore, the levels of cytokines and chemokines can serve as a measure of disease progression, for example, elevated levels of CXCL10 (IP-10) in the peripheral blood during primary HIV infection appears to be a sign of rapid disease progression (*Ploquin et al., 2016*). In fact, permissiveness to HIV infection by different CD4+ T-cell subsets based on the differential expression of CXCR3, CCR4 and CCR6 has been clearly elucidated (*Gosselin et al., 2010*; *Gosselin et al., 2017*). Host defense also includes the release of effector cytokines and lytic proteins, such as perforin and granzyme B by CD4+ as well as CD8+ T cells for killing virally infected cells.

Complement represents a key component of the innate defense armory and the initial host-HIV interaction also includes the activation of complement system with all three cascade pathways (*Stoiber et al., 2008*). Upon infection, hydrolysis of C3 into C3b occurs on the viral surface, which sequentially becomes inactivated into iC3b (*Stoiber et al., 2008*). The C1q pathway is also activated and C1q fixation to the virions are especially efficient in the presence of HIV-1 gp120 and gp41 specific antibodies (*Prohászka et al., 1999*). The virions transferred during sexual transmission via vaginal and rectal intercourse should be complement and/or complement antibody opsonized particles (*Stoiber et al., 2008*). The complement components are recognized by different receptors, especially complement receptors (CRs) expressed on various cell types (*Stoiber et al., 2008*). Rectal mucosa is known to house CR3 (CD11b/CD18) on epithelial cells, DCs, and monocytes/macrophages, which have affinity towards iC3b as evident from studies using immune cells from blood (*Bouhlal et al., 2001*), a colorectal cell line (*Bouhlal et al., 2002*), or tissue biopsies (*Hussain et al., 1995*). It is intriguing that complement activation and opsonization renders HIV more accessible to host cells. Indeed, our previous observations have shown that not only were complement-opsonized HIV internalized more efficiently by DCs (*Tjomsland et al., 2011*), but also that opsonization decreased inflammatory and antiviral responses to enhance infection of immature DCs as compared to free HIV (*Ellegård et al., 2015*; *Ellegård et al., 2014*). So far only few studies exist examining the ex vivo HIV-1 infection of human colorectal tissue from HIV negative individuals (*Fletcher et al., 2006*; *Herrera et al., 2011*; *Kordy et al., 2018*; *Abner et al., 2005*), none as far as we are aware of that examine the initial effects exerted by HIV exposure on the colorectal mucosa. In this study we have explored effects exerted by free and opsonized virus on the initial responses in colorectal tissue explants and the effect HIV exposure had on emigrated and isolated immune cells from colorectal tissue and the subsequent responses generated by the HIV exposed cells.

Our results in the current study indicated that HIV opsonization altered the transcriptome profiles, with the free virus activating different innate and inflammatory pathways during early stages of infection compared to complement opsonized HIV, for example induction of type I IFN, TLR signaling, and Th17 responses. Noteworthy, at 96 hr the complement-opsonized HIV had created an environment with higher inflammatory responses, allowing for higher levels of HIV infected immune cells. In conclusion, opsonization of HIV alters the activation of signaling pathways and cells in the colorectal mucosa in a manner that promotes viral establishment and by creating an environment that stimulates mucosal T cell activation and inflammatory T helper cells, and this could play an important role in HIV immunopathogenesis.

## Results

### HIV-induced activation of innate and inflammatory pathways in colorectal mucosa

The effect HIV infection exerts on the GI mucosa is detrimental, with high levels of infection and destruction of CD4+ T cells and inflammation (*Brenchley and Douek, 2008*; *Mait-Kaufman et al., 2015*). Given the scarcity of data on the early stages of mucosal HIV infection and the effect it has on colorectal tissues, we set out to investigate the early events of infection of free HIV (F-HIV), complement opsonized HIV (C-HIV), and complement and antibody opsonized HIV (CI-HIV) at the transcriptomics level after 24 hr of virus exposure. The tissue biopsies obtained from surgery were subjected to histopathological examination by the clinical pathologist and showed no histological signs of malignancy, infection, or inflammation (*Figure 1—figure supplement 1*) and the complement and IgG opsonization verified by ELISA (*Figure 1—figure supplement 2A–C*). The number of significantly upregulated genes after 24 hr of HIV exposure, assessed in IPA, was highest in F-HIV, whereas when assessing 3-fold, 5-fold and 10-fold upregulated genes, the levels were similar across the study groups, that is F-HIV, C-HIV and complement and CI-HIV (*Figure 1A*). We found that an array of canonical pathways was affected by HIV exposure, and type I IFNs and antiviral and PRR pathways were highly upregulated in the colorectal mucosa following exposure to F-HIV but remained low or unaltered in the C-HIV-exposed tissues (*Figure 1B*). In addition, there was clear evidence of activation of IL-17-related pathways by F-HIV. Of note, complement opsonization of HIV led to the suppression of several T helper cell costimulatory signaling pathways (*Figure 1B*). Gene ontology (GO) analysis also revealed similar effects as the canonical pathways with type I IFN

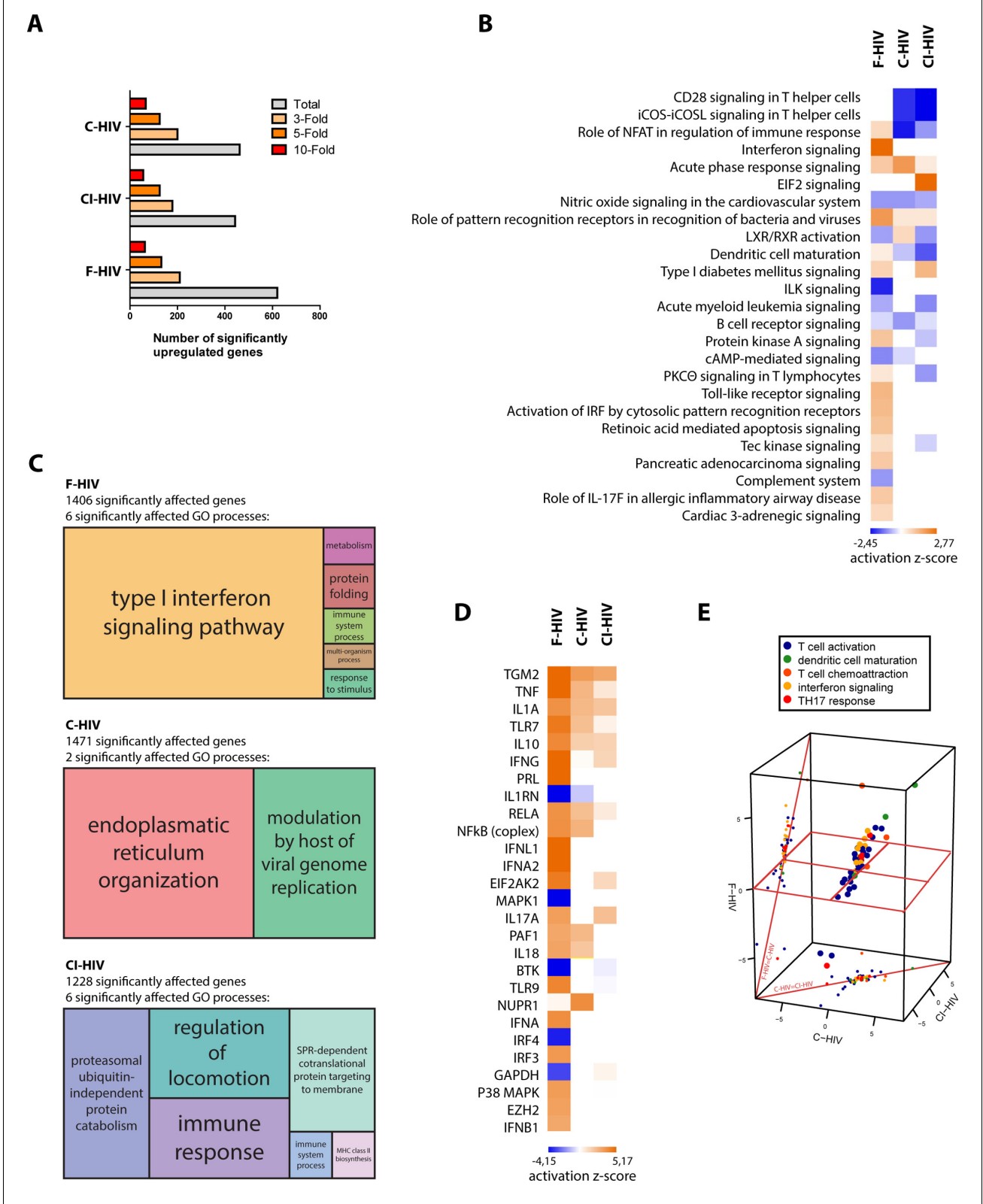

**Figure 1.** HIV-induced activation of innate and inflammatory pathways in colorectal mucosa with high level of antiviral factors in F-HIV. The colorectal tissue biopsies were exposed to HIV-1$_{BaL}$ (250 ng/mL), either free (F-HIV), complement-opsonized (C-HIV), or virions opsonized by a cocktail of complement and antibodies (CI-HIV) or mock-treated, by spinning the cultures for 2 hr at 37°C. The tissues were washed and transferred to six-well plates and cultured for 24 hr. The tissues and emigrated cells were harvested and whole transcriptome sequencing was performed. (**A**) Analyses of

*Figure 1 continued on next page*

*Figure 1 continued*

amount of significantly upregulated or down regulated genes assessing total, 3-fold, 5-fold, and 10-fold of upregulated changes as compared to mock. (B) Canonical pathways affected by the HIV exposure were assessed by IPA and presented as a heat map with the threshold for p-values set to log 1.3 and presented as activation Z-score and with a Z-score cutoff of 3. (C) Gene enrichment analysis was done in R with an algorithm from REVIGO where all fold changes were ranked of genes affected with p<0.05. (D) Analysis of upstream regulators affected by the HIV exposure was assessed by IPA and presented as a heat map with an activation Z score of 3 or higher. (E) The top upstream regulators were divided into functional groups and a 3D scatterplot was used to visualize clustering. The division of genes after function was down in IPA including only genes with p<0.05 and the 3D scatterplot is made in plat3D in R (N = 6). *p<0.05, **p<0.001, ***p<0.0001. Data is shown as mean ± SEM.

The online version of this article includes the following figure supplement(s) for figure 1:

**Figure supplement 1.** Histological images of colonic mucosa.
**Figure supplement 2.** Complement and IgG opsonization of HIV-1.
**Figure supplement 3.** Complement opsonization elevated the antiviral and HIV regulatory factors in colorectal tissue.

signaling pathway as the major statistically significantly affected GO process in F-HIV followed by protein folding, immune system processes, metabolism, response to stimuli and multi-organism process. C-HIV significantly affected two major GO processes, which were modulation by host of viral genome and endoplasmic reticulum organization. Immune response and regulation of locomotion were the main affected GO processes in CI-HIV-exposed mucosal tissues, and in addition ubiquitin-independent protein catabolism, SPR-dependent co-translational protein targeting to membrane, immune system process and MHC class II biosynthesis were also enriched processes (*Figure 1C*). In summary, exposure to free HIV induced the activation of innate and inflammatory pathways, including Th17 related pathways in the colorectal mucosa. Complement opsonization of the virus, on the other hand, suppressed these pathways.

## Free HIV-induced high level of antiviral upstream regulators

Next, we analyzed the upstream antiviral regulators assessing a heat map sorted by Z-Score and found a clear pattern with type I IFNs namely IFN-α1, IFN-λ and IFN-β1, P38 MAPK and IRF3 that were significantly upregulated for F-HIV but remained unaffected in the C-HIV groups (*Figure 1D*). The inflammatory responses as measured by the expression of IL-1α, IL-1β and TNF were strongest for F-HIV but were also seen although at lower levels in the complement opsonized HIV groups (*Figure 1D*). Of note, the gene TGM2, activated by retinoic acid, was highly expressed across all the three HIV-exposed groups (*Figure 1D*). Furthermore, the top upstream regulators were divided into functional groups, and when a 3D PCA scatterplot was used to visualize clustering, we found evidence of interferon signaling, Th17 responses and markers of T-cell activation standing out as factors separating the different HIV-exposed groups, F-HIV resulted in a stronger activation of these responses than with C-HIV (*Figure 1E*). Some of the factors that we found in the transcriptome analysis to be significantly affected in the colorectal tissue exposed to F-HIV, C-HIV and CI-HIV were followed up by mRNA level analysis by PCR of immune cells isolated from colorectal mucosa for some antiviral, inflammatory, and HIV regulatory factors (*Ellegård et al., 2014*). At 24 hr, we found significantly increased levels of MXA (p=0.033), IFIT3 (p=0.0029), APOBEC3A (p=0.022), tetherin (p=0.012), SAMHD1 (p=0.022), and TREX (p=0.038) in F-HIV exposed mucosal cells as compared to unexposed cells (*Figure 1*, *Figure 1—figure supplement 3A-B*). Compared to F-HIV there were significantly lower levels in C-HIV and CI-HIV of IFIT3 (p=0.046, p=0.016) and SAMHD1 (p=0.036, p=0.039), and of TREX (p=0.015) in C-HIV exposed mucosal cells (*Figure 1 Figure 1—figure supplement 3A-B*). In addition, APOBEC3A and tetherin transcripts were also upregulated for C-HIV and CI-HIV, with significantly higher levels for C-HIV of both APOBEC3A (p=0.022) and tetherin (p=0.0013), and CI-HIV for tetherin (p=0.008) (*Figure 1 Figure 1—figure supplement 3A-B*). Of note, there was a variation in the mRNA levels induced by HIV exposure between the different donors, but the general profiles were similar. This could be due to donor-specific differences in expression kinetics of the response to HIV-1, i.e. the peak mRNA expression levels occurred at some earlier time-points and had subsided by 24 hr in some samples (*Ellegård et al., 2014*; *Ellegård et al., 2018*). Taken together, the free HIV induced high levels of antiviral upstream regulators in the colorectal mucosa that remained unaffected when the virions were complement-opsonized, whereas inflammatory factors were upregulated in all groups albeit to higher levels by free HIV.

## Complement opsonization elevated the levels of HIV-infected colorectal mucosal immune cells

Given that little is known about the initial infection, and the effect of HIV exposure on colorectal mucosal immune cells, we next studied the infection levels in primary mucosal immune cells to see if we could see connections to the immune profiles and effect on CD4+ and CD8+ T cells. To explore the mucosal immune cells in an efficient manner the colorectal mucosa immune cells were isolated through enzymatic tissue disruption and to ensure that this treatment did not affect the HIV infection we also assessed the infection in cells emigrating out of colorectal tissue explants. The levels of HIV infection in mucosal DCs and T cells emigrating out of colorectal tissue explants, exposed to mock, F-HIV, C-HIV or CI-HIV were assessed for HIV-1 p24 expression by flow cytometry at day 4–5. Complement opsonization significantly increased HIV infection of emigrated mucosal DCs relative to F-HIV with a 2.9 fold increase for C-HIV (p=0.0156) and 2.6 fold increase for CI-HIV (p=0.017) (*Figure 2A*), although no such effect was evident in the emigrated T cells (*Figure 2B*). The isolated immune cells were exposed to the same HIV conditions as the tissue biopsies and we assessed HIV-1 infection in DC, CD163+ macrophages and CD4+ T cells, which should represent the immune cells susceptible to HIV-1 infection. The infection profiles of mucosal DCs isolated from the colorectal tissue matched the emigrated DCs, with complement opsonization giving rise to significantly higher levels of HIV infection with a 2.9 fold increase for C-HIV (p=0.0012) and 4.5 fold increase CI-HIV (p=0.0006) (*Figure 2A,C–D*). The increase in infection by C-HIV with a two fold increase (p=0.009) and CI-HIV with a 1.95 fold increase (p=0.012) compared to F-HIV, was also seen for the CD163+ macrophages in the isolated mucosal immune cells (*Figure 2E–F*). In the case of isolated mucosal CD4 T cells, complement opsonization did not result in statistically significant changes in HIV infection among the isolated mucosal CD4+ T cells as compared to F-HIV (*Figure 2B,G–H*). (See *Figure 2—figure supplement 1* for gating strategies). To get an overall view of the infection profile of all infected immune cells, i.e. DCs, macrophages and CD4+ T cells, we pooled the data, analyzed, and found that the C-HIV and CI-HIV gave a significant fold increase in infection compared to F-HIV (*Figure 2I*). These findings indicate that the complement opsonization influenced the infection profiles with increased levels of infection with C-HIV and CI-HIV as compared to F-HIV. The enhanced infection should not be due to altered viral tropism (*Tjomsland et al., 2013a*), rather more efficient infection that could depend on access to other receptors facilitating the viral binding and induction of different signaling pathways (see *Figure 1*) that support the higher infection.

## Exposure to HIV affected the composition of mucosal CD8+ memory T cell subsets

HIV infection is known to influence aspects of the T cell differentiation both in the blood (*Tanko et al., 2018*) as well as in the draining lymph nodes either by direct effects on the infected cells or by bystander effects on the uninfected cells (*Buggert et al., 2018*). Here, we explored if mere exposure to HIV was enough for the modulation of memory CD4+ and CD8+ T cell subsets isolated from colorectal mucosa (See *Figure 3—figure supplement 1A–B* for gating strategies and CD3 T cell viability). We assessed CD45RA+CCR7+ (naïve), CD45RA-CCR7+ (central memory), CD45RA$^-$CCR7$^-$ (effector memory) and CD45RA$^+$CCR7$^-$ (terminal effector) subsets 96 hr after exposure to HIV (*Figure 3A–B*). The vast majority of isolated mock treated CD4+ T cells displayed effector memory phenotypes and the HIV exposure did not significantly alter the proportions of memory subsets (*Figure 3A*). In addition, the CD8+ T-cell populations were also predominately made up by the effector memory subsets (*Figure 3B*). However, upon exposure to HIV (F-HIV, C-HIV and CI-HIV) the frequencies of naïve CD8+ T cells decreased from 8.2% for mock, to 6.7% for F-HIV, 2.3% (p=0.01) for C-HIV, and 3.9% (p=0.041) for CI-HIV (*Figure 3B*).

## Role of HIV exposure on regulation and differentiation T helper cell subsets

Enrichment of pathways associated with TH17 activation in colorectal tissue upon HIV exposure (*Figure 1*) lead us to analyze mRNA after 24 hr HIV exposure (*Figure 4—figure supplement 1*) and protein expression after 96 hr HIV exposure (*Figure 4A–D*) of transcription factors known to regulate principal helper T cell subsets. This included factors associated with Th1; Tbet and IFN-γ, Th2; GATA3 and IL-5, Th17; RORγt, IL-17A, and IL-23, and factors associated with T cell suppression/

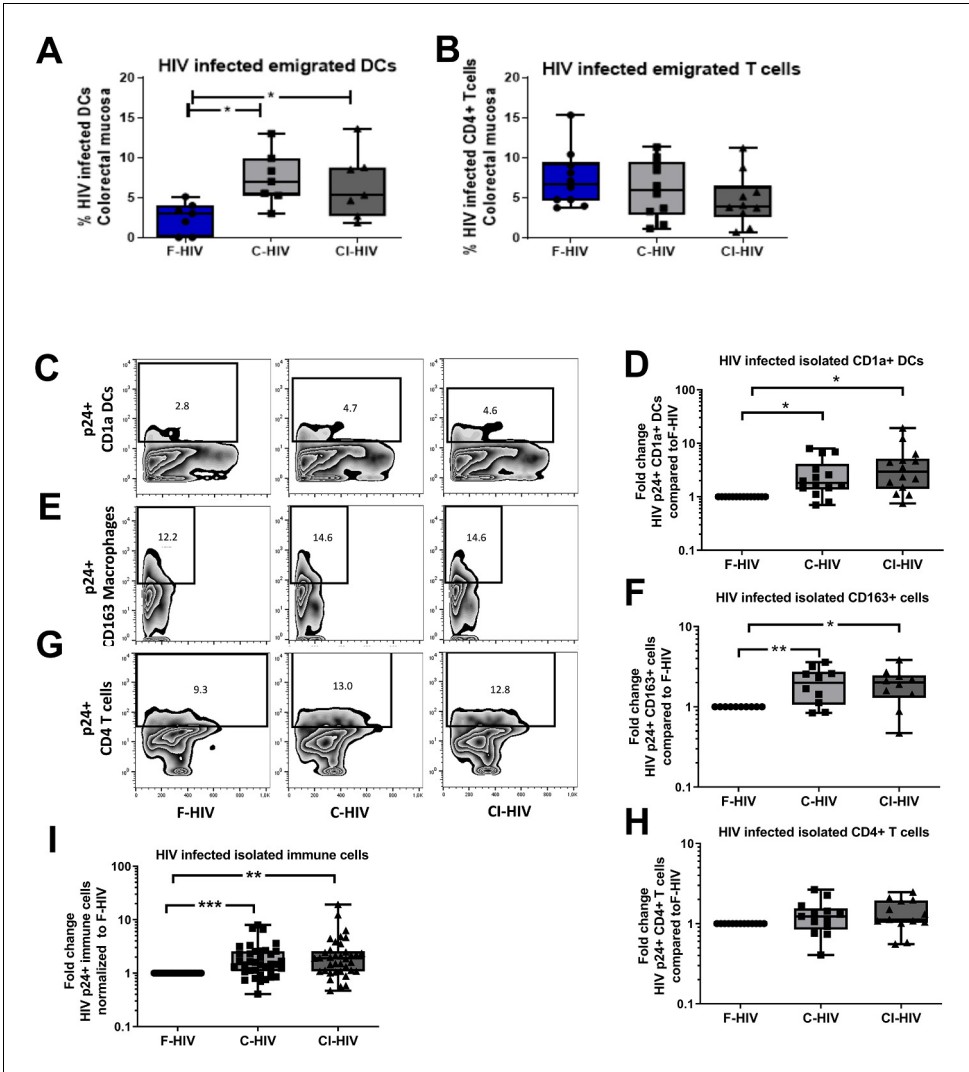

**Figure 2.** Complement opsonization elevated the levels of HIV-infected colorectal mucosal immune cells. The colorectal tissue biopsies were exposed to HIV-1$_{BaL}$ (250 ng/mL), either free (F-HIV), complement-opsonized (C-HIV), or virions opsonized by a cocktail of complement and antibodies (CI-HIV) or mock-treated, by spinning the cultures for 2 hr at 37°C. The tissues were washed and transferred to six-well plates and cultured for 4–5 days. (**A, B**) The emigrating cells were harvested and stained with (**A**) anti-CD1a and anti-HIV-1 mAbs for DCs, and (**B**) anti-CD3, anti-CD4, and anti-HIV-1 mAbs for CD4+ T cells and the level of infection was assessed by flow cytometry (N = 7–8). (**C–G**) Mucosal immune cells isolated from the colorectal tissue biopsies by enzymatic digestion were exposed to d HIV-1$_{BaL}$, either free (F-HIV), complement-opsonized (C-HIV), or virions opsonized by a cocktail of complement and antibodies (CI-HIV) or mock-treated, by spinning the cultures. The level of HIV infection (**C, E, and G**) and fold change of HIV infected cells compared to F-HIV infection levels (**D, F, and H**) were assessed after 4 days. The immune cells were stained with (**C–D**) anti-CD1a and anti-HIV-1 mAbs for DCs, (**E–F**) anti-CD163 and anti-HIV-1 mAbs for macrophages (**G–H**), and anti-CD3, anti-CD4, and anti-HIV-1 mAbs for CD4+ T cells and the level of infection was assessed by flow cytometry (N = 11–15). The fold change of all HIV infected immune cells combined compared to F-HIV infection levels was assessed after 4 days (**I**). Statistical significance was tested using repeated measures of ANOVA followed by Tukey's posttest (**A–B**) and One-tailed Paired T-test (**D, F, H, and I**). *p<0.05, **p<0.01, ***p<0.001. Data are shown as mean ± SEM. Zebra plots of (**C**) CD1a-DC, (**E**) CD163-Macrophages and (**G**) CD4-T cells are representative of one individual. DCs, Macrophages and T cells infected with HIV were identified based on percentage populations of CD1a+p24+, CD163+p24+ and CD4+p24+ cells, respectively.

The online version of this article includes the following figure supplement(s) for figure 2:

**Figure supplement 1.** Representative Zebra plots with gating for HIV-1 p24+ Immune cells.

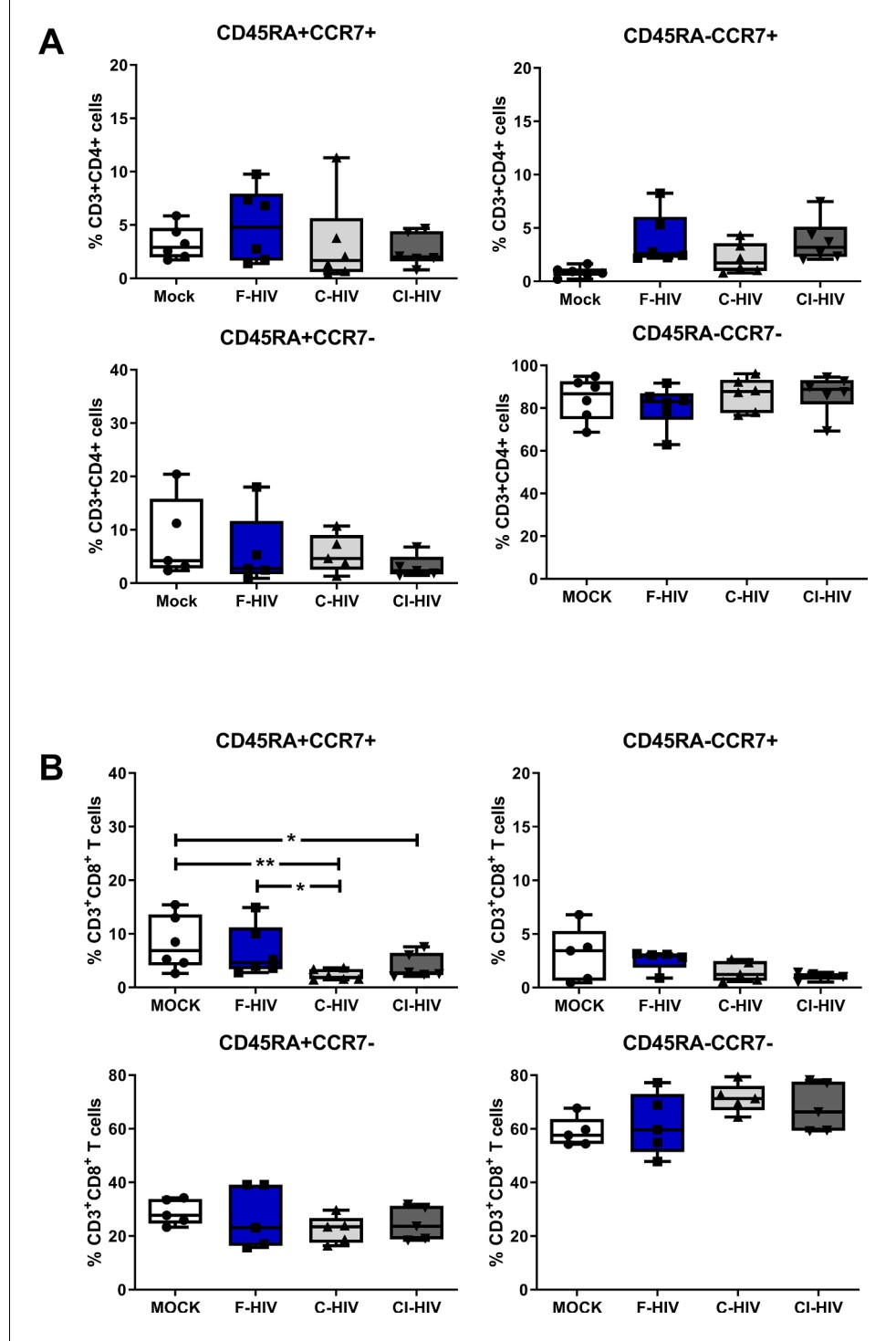

**Figure 3.** Alteration in mucosal CD4+ and CD8+ T cell memory cell after HIV-1 exposure. Mucosal immune cells isolated from the colorectal tissue biopsies by enzymatic digestion were exposed to HIV-1$_{BaL}$ (250 ng/mL), either free (F-HIV), complement-opsonized (C-HIV), or virions opsonized by a cocktail of complement and antibodies (CI-HIV) or mock-treated, by spinning the cultures. The effect HIV exposure had on mucosal T cells phenotype was assessed after 4 days. (A–B) The immune cells were stained with CD3, CD4, CD8, CD45RA, and CCR7 mAbs. Percentage of CD45RA+CCR7+, CD45RA+CCR7-, CD45RA-CCR7+, and CD45RA-CCR7- expressing CD4+CD3+ T cells (A) or CD8+CD3+T cells (B) unexposed or exposed to F-HIV, C-HIV and CI-HIV was assessed by flow

*Figure 3 continued on next page*

*Figure 3 continued*

cytometry (N = 5–6). Statistical significance was tested using repeated measures of ANOVA followed by Tukey's posttest. *p<0.05, **p<0.01, ***p<0.001. Data are shown as mean ± SEM.

The online version of this article includes the following figure supplement(s) for figure 3:

**Figure supplement 1.** The gating strategy for the CD4+ and CD8+ T cell subsets and the viability of the CD3+ T cells.

regulatory T cells (Treg); FoxP3 and IL-10. The frequency of CD4+ T cells and CD8+ cells expressing the transcription factors were measured by flow cytometry in immune cells isolated from colorectal tissues at 96 hr (*Figure 4A–D*). In addition, following a recent report (*Sun et al., 2015*) involving susceptibility of Th1/Th17 cells to HIV infection, we also assessed the double-positive RORγt+Tbet+ colorectal cells. Tbet (mean 34.7% vs 19.5%, p=0.001), GATA3 (mean 49% vs 32%, p=0.007) and RORγt (mean 72.2% vs 46%, p=0.003) expressing CD4+ T cells were significantly increased in F-HIV-exposed cells compared to unexposed cells (*Figure 4A*). The levels of CD4+ T cells expressing RORγt or GATA3 remained unaffected by C-HIV exposure whereas the Tbet expression increased (mean 30.5% vs 19.5%, p=0.022), but to a lesser extent than F-HIV-exposure (*Figure 4A*). The level of RORγt+CD4+ T cells expressing Tbet increased significantly after exposure to F-HIV (mean 47.3% vs 35%, p=0.006), C-HIV (mean 57% vs 35%, p=0.001) and CI-HIV (mean 47.7% vs 35%, p=0.032) relative to unexposed cells (*Figure 4B*). Within the CD8+ population there were a decline in frequencies of cells expressing Tbet or RORγt after exposure to C-HIV (Tbet mean 27.5% vs 38.6%, p=0.027 and RORγt mean 52.1% vs 68.5%, p=0.0019) and CI-HIV (Tbet mean 27.2% vs 38.6%, p=0.021 and RORγt mean 54.2% vs 68.5%, p=0.0076) compared to F-HIV ('*Figure 4C*), whereas the expression levels of GATA3 and FoxP3 were not affected by any form of HIV exposure (*Figure 4C*). The levels of RORγt+CD8+ T cells expressing Tbet were similar for HIV-exposed and unexposed isolated mucosal T cells (*Figure 4D*). The mRNA profiles were similar to the protein profiles (*Figure 4* and *Figure 4—figure supplement 1A-D*). In summary, the HIV exposure tended to skew CD4+ T cells towards Th1Th17 and complement opsonization of the virus was associated with decreased expression of Tbet and RORγt within the CD8+ T cell compartment.

## HIV infection increases the amount of CD4+ T cell expressing CCR4 +CCR6+ and CXCR3+CCR6+

Pursuing the observation regarding the effects of HIV on different T cell subsets at the transcription level and the importance of functional T cell migration, we next investigated the impact of HIV on the expression pattern of chemokine receptors (CCR4, CCR6 and CXCR3), i.e. CXCR3+CCR6- (Th1), CCR4+CCR6- (Th2), CCR4+CCR6+ (Th17) and CXCR3+CCR6+ (Th1Th17) subsets (*Gosselin et al., 2010*; *Figure 4E,F*). It has to be noted that CCR4+CCR6+ and CXCR3+CCR6+ CD4+ T cells are highly susceptible to HIV infection (*Gosselin et al., 2010*). The proportion of each of the four different T cell subsets not exposed to HIV were: Th1 10% (mean value), Th2 7%, Th17 30%, and Th1Th17 53% (*Figure 4E*). F-HIV (mean 33% vs 16.2%, p=0.0076), C-HIV (mean 30% vs 16.2%, p=0.032) and CI-HIV (mean 34% vs 16.2%, p=0.0056) exposure significantly elevated the frequencies of CD4+ T cells with Th17 phenotype, while the difference was significant for F-HIV (mean 63% vs 28%, p=0.0005) and CI-HIV (mean 51% vs 28%, p=0.029) in case of Th1Th17 subsets relative to unexposed cells (*Figure 4F*). The frequencies of CD4+ T cells with Th1 and Th2 phenotypes were not affected by HIV exposure (*Figure 4F*). Clearly HIV exposure increased the amounts of isolated colorectal mucosal CD4+ T cell subsets that are susceptible to infection by HIV (*Mehandru et al., 2004*; *Brenchley et al., 2004*; *Gosselin et al., 2010*).

## HIV induces modulation of immune checkpoint molecules on CD4+ helper T cells and CD8+ T cells

The thresholds for T-cell activation are dependent on the location of cells, for instance, blood cells normally have lower thresholds than cells present in tissues that are under constant exposure to pathogens and indigenous microbiota (*Hu and Pasare, 2013*; *van Wijk and Cheroutre, 2010*). Considering the activation threshold for T cells is likely influenced by the expression of an array of negative and positive immune checkpoint factors, we analyzed the effect HIV had on the expression of

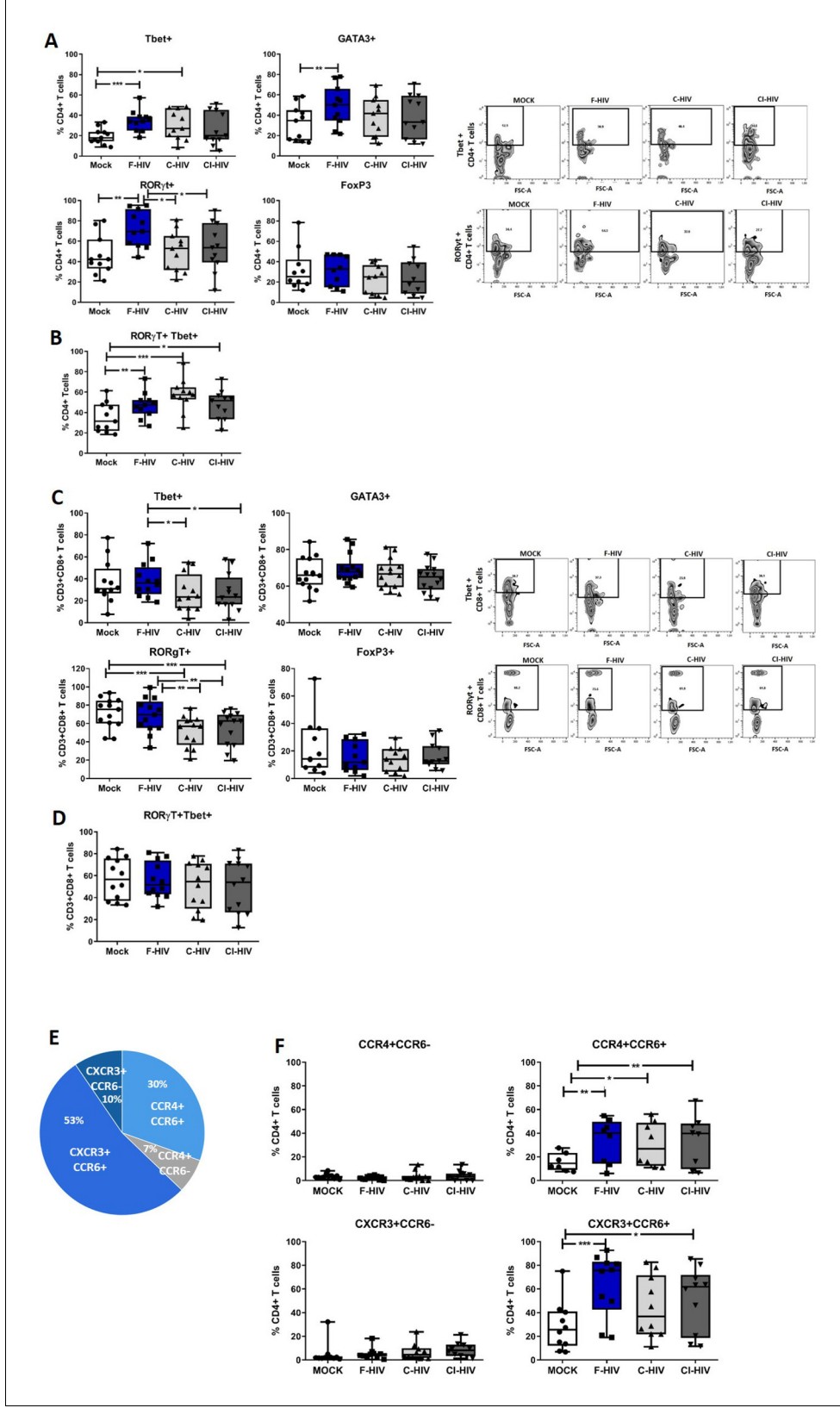

**Figure 4.** HIV-1 exposure alters the T cell expression of transcriptional factors and chemokine receptors. Mucosal immune cells isolated from the colorectal tissue biopsies by enzymatic digestion were exposed to HIV-1$_{BaL}$ (250 ng/mL), either free (F-HIV), complement-opsonized (C-HIV), or virions opsonized by a cocktail of complement and

*Figure 4 continued on next page*

*Figure 4 continued*

antibodies (CI-HIV) or mock-treated, by spinning the cultures. The effect HIV exposure had on mucosal T cells phenotype was assessed after 4 days. (A–D) The immune cells were stained with CD3, CD4, CD8, Tbet, GATA3, RORγt, and FoxP3 mAbs. (A–B) Percentage of unexposed, F-HIV, C-HIV, or CI-HIV exposed CD4+ T cells expressing GATA3, Tbet, RORγt, or FoxP3, or RORγt and Tbet were assessed by flow cytometry. (C–D) Percentage of unexposed, F-HIV, C-HIV, or CI-HIV exposed CD8+ T cells expressing GATA3, Tbet, RORγt, or FoxP3, or RORγt and Tbet were assessed by flow cytometry. (E–F) The immune cells were stained with CD3, CD4, CD8, CCR4, CCR6, and CXCR3 mAbs. (E) The proportion of each of the four T helper populations: CCR4+CCR6-, CCR4+CCR6+, CXCR3+CCR6- or CXCR3+CCR6+ among unexposed CD4+ T cells were assessed by flow cytometry. (F) Percentage of unexposed, F-HIV, C-HIV, or CI-HIV exposed CD4+ T cells expressing CCR4+CCR6-, CCR4+CCR6+, CXCR3+CCR6- or CXCR3+CCR6+ were assessed by flow cytometry (N = 12–15). Statistical significance was tested using repeated measures of ANOVA followed by Tukey's posttest. *p<0.05, **p<0.01, ***p<0.001. Data are shown as mean ± SEM.

The online version of this article includes the following figure supplement(s) for figure 4:

**Figure supplement 1.** Exposure to HIV modulated the transcription factor and cytokines associated with different T-cell subsets in the isolated mucosal immune cells.

some of the major immune checkpoint molecules like PD-1, TIM3, LAG3 and CD160 on isolated mucosal CD4+ T cells and CD8+ T cells (*Figure 5*). The extent of CD4+ T cells expressing the PD-1 (mean 47.6% vs 28.9%, p=0.008), TIM3 (mean 59.3% vs 32.2%, p=0.0011), LAG3 (mean 65.8% vs 39.1%, p=0.002) and CD160 (mean 73.2% vs 50.3%, p=0.0095) coinhibitory molecules was significantly elevated after exposure to F-HIV compared to unexposed cells (mock) (*Figure 5A*). Moreover, for CD4+ T cells expressing PD-1 (mean 35.6% vs 47.6%, p=0.04), LAG3 (mean 52.9% vs 65.8%, p=0.035) and CD160 (mean 56% vs 73.2%, p=0.034), a significant difference was observed between F-HIV and C-HIV treatments (*Figure 5A* and *Figure 5—figure supplement 1* for representative Zebra plots). The levels of CD8+ T cells expressing immune checkpoint molecules remained either unaffected or lowered when exposed to complement opsonized HIV, with significantly smaller number of cells co-expressing PD-1 (mean 38.7% vs 29.6%, p=0.0026: mean 38.7% vs 26.5%, p=0.0005) and LAG3 (mean 55.7% vs 36.7%, p=0.009: mean 55.7% vs 29.4%, p=0.0005) for both C-HIV and CI-HIV, and with CD160 (mean 46.9% vs 35%, p=0.0019) for CI-HIV-exposed cells (*Figure 5B* and *Figure 5—figure supplement 1* for representative Zebra plots). There were no or little effects of F-HIV on the CD8+ T cell expression of immune checkpoint factors besides for LAG3, which was significantly decreased (mean 55.7% vs 38.1%, p=0.036) compared to mock (*Figure 5B*). The CD8+CD38+ cell population was also included in the analysis given that CD38 appears to play an active role in chronic HIV infection, where increased expression of CD38 on CD8+ T cells has a clear association with immune activation and HIV disease progression (*Hoffmann et al., 2016*; *Klatt et al., 2013b*). The frequency of CD38+ CD8+ T cells almost remained unaltered by HIV exposure, but a significant fraction of the CD38+ population lost the expression of PD-1 (mean 60.3% vs 47.9 .%, p=0.0011: mean 60.3% vs 42%, p=0.0007) when exposed to C-HIV or CI-HIV and CD160 (mean 60.3% vs 49.6 .%, p=0.0044: mean 60.3% vs 47.9 .%, p=0.0011: mean 60.3% vs 42%, p=0.0007) when exposed to F-HIV, C-HIV or CI-HIV (*Figure 5C*). Our data show that HIV exposure altered the amount of mucosal T cells that expressed immune checkpoint molecules such as PD-1 and LAG3.

## HIV exposure modulated transcriptional regulation of mucosal T cells

The identifying features of colorectal mucosal T cells along with expression of different immune checkpoint molecules include the transcription factors EOMES and Tbet, which, in turn, are responsible for maintaining the population (*Wherry and Kurachi, 2015*). Although the number and expression pattern of Tbet and EOMES differed extensively between resting CD4+ and CD8+ T cells, their constitutive expression is known to delineate specific cellular T-cell subsets in blood (*Knox et al., 2014*). In addition, EOMES plays an important role in regulating the cytotoxic functions, development and survival, and EOMES expression appears to increase when T cells differentiate into memory cells, which however, is not the case for Tbet (*McLane et al., 2013*; *Pearce et al., 2003*). We analyzed both single and co-expression of Tbet and EOMES in association with PD-1, LAG3, TIM3 and CD160 on CD4+ and CD8+ T cells isolated from colorectal tissue to underline their role in HIV infection (*Figure 6*). Tbet+CD4+ T cells had a significant increased expression of PD-1 (mean 35.3%

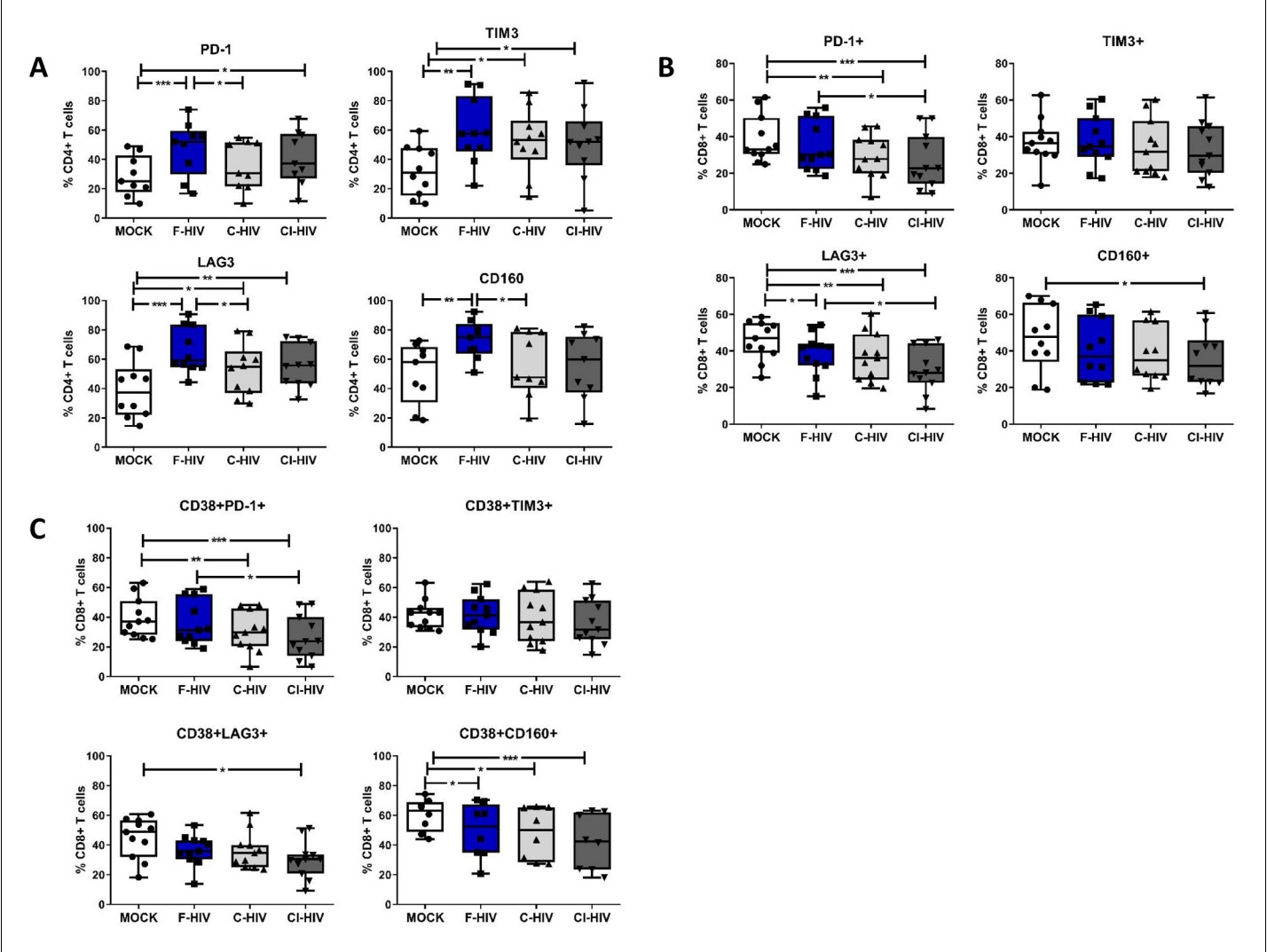

**Figure 5.** HIV-1 exposure alters the levels of negative immune checkpoint factors on the mucosal T cells. Mucosal immune cells isolated from the colorectal tissue biopsies by enzymatic digestion were exposed to HIV-1$_{BaL}$ (250 ng/mL), either free (F-HIV), complement-opsonized (C-HIV), or virions opsonized by a cocktail of complement and antibodies (CI-HIV) or mock-treated, by spinning the cultures. The effect HIV exposure had on mucosal T cells phenotype was assessed after 4 days. (A–C) The immune cells were stained with CD3, CD4, CD8, CD38, PD-1, LAG3, TIM3, and CD160 mAbs. (A) Percentage of unexposed, F-HIV, C-HIV, or CI-HIV exposed CD4+ T cells expressing PD-1, TIM3, LAG3 or CD160+ CD4+ T cells, (B) Percentage of unexposed, F-HIV, C-HIV, or CI-HIV exposed CD8+ T cells expressing PD-1, TIM3, LAG3 or CD160, and (C) Percentage of unexposed, F-HIV, C-HIV, or CI-HIV exposed CD38+CD8+ T cells expressing PD-1, TIM3, LAG3 or CD160 were assessed by flow cytometry (N = 12–15). Statistical significance was tested using repeated measures of ANOVA followed by Tukey's posttest. *p<0.05, **p<0.01, ***p<0.001. Data are shown as mean ± SEM.

The online version of this article includes the following figure supplement(s) for figure 5:

**Figure supplement 1.** Representative Zebra plots for immune checkpoint factors PD1 and LAG3 on mucosal T cells.

vs 53.6%, p=0.008), LAG3 (mean 45.7% vs 72.5%, p=0.0003) and CD160 (mean 56.8% vs 76.9%, p=0.0032) when exposed to F-HIV as compared to unexposed cells (*Figure 6A*). However, the levels of Tbet+CD4+ T cells expressing PD-1 (mean 53.6% vs 40.6%, p=0.028) or LAG3 (mean 72.5% vs 57.9%, p=0.026) were significantly decreased after C-HIV compared to F-HIV (*Figure 6A*). The levels of CD160, although not significant, followed the same trend. A representative viSNE plot based on clustering of PD-1+Tbet+ on CD4 T cells illustrated the difference between F-HIV and C-HIV conditions (*Figure 6A*). The levels of PD-1 and LAG3 expressing Tbet+EOMES+CD4+ T cells were increased in F-HIV-exposed mucosal immune cells (*Figure 6B*).

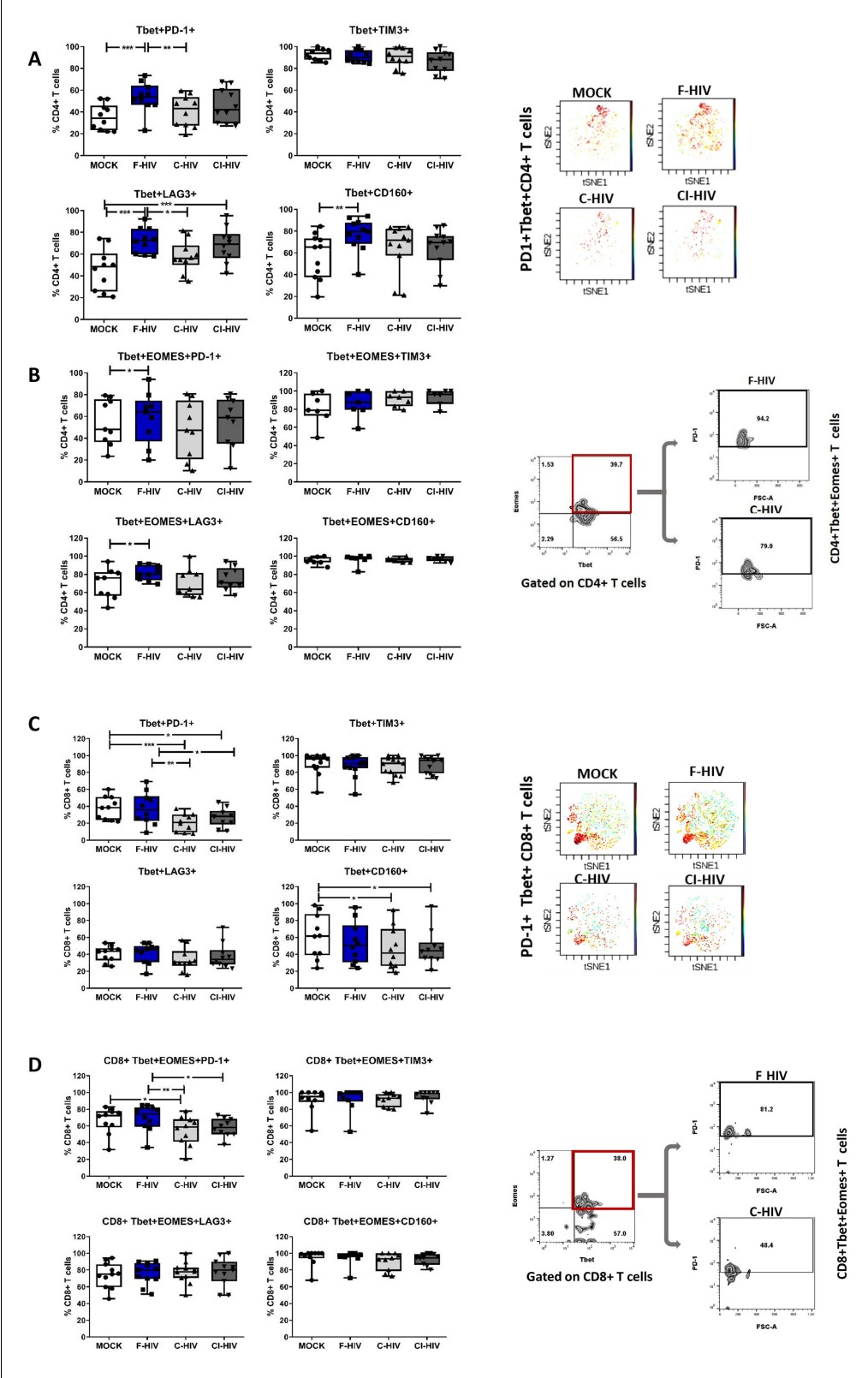

**Figure 6.** HIV exposure modulated transcriptional regulation of mucosal T cells. Mucosal immune cells isolated from the colorectal tissue biopsies by enzymatic digestion were exposed to HIV-1$_{BaL}$ (250 ng/mL), either free (F-HIV), complement-opsonized (C-HIV), or virions opsonized by a cocktail of complement and antibodies (CI-HIV) or mock-treated, by spinning the cultures. The effect HIV exposure had on mucosal T cells phenotype was assessed after 4 days. (A–D) The immune cells were stained with CD3, CD4, CD8, Tbet, EOMES, PD-1, LAG3, TIM3, and CD160 mAbs. (A) Percentage

*Figure 6 continued on next page*

Figure 6 continued

of unexposed, F-HIV, C-HIV, or CI-HIV exposed Tbet+CD4+ T cells expressing PD-1, TIM3, LAG3 or CD160. viSNE plot of PD-1+Tbet+CD4+ was constructed on CD4 T cells, data presented as dot plots with colored channels. Data from a donor shows CD4 T cells, subjected to tSNE algorithm, which provides cells with a unique coordinate according to its expression of PD-1+T-bet+ parameters, displayed on a two-dimensional plot (tSNE1 versus tSNE2). The heat gradient (blue to red) indicates expression level of Tbet+PD-1+ on CD4 following exposure to different conditions of viruses. (B) Percentage of unexposed, F-HIV, C-HIV, or CI-HIV exposed Tbet+EOMES+ CD4+ T cells expressing PD-1, TIM3, LAG3 or CD160 were assessed by flow cytometry. Flow cytometry Zebra plots for Tbet+EOMES+ CD4+ T cells expressing PD-1. (C) Percentage of Tbet+CD8+ T cells expressing PD-1, TIM3, LAG3 or CD160. viSNE plot of PD-1+Tbet+CD8+ was constructed on CD8 T cells, and data are presented as dot plots with colored channels. Data from a donor shows CD8 T cells, subjected to tSNE algorithm, which provides cells with a unique coordinate according to its expression of PD-1 +T-bet+ parameters, displayed on a two-dimensional plot (tSNE1 versus tSNE2). The heat gradient (blue to red) indicates expression level of Tbet+PD-1+ on CD8 following exposure to different conditions of viruses. (D) Percentage of unexposed, F-HIV, C-HIV, or CI-HIV exposed Tbet+EOMES+ CD4+ T cells and percentage of Tbet+EOMES+ CD4+ T cells expressing PD-1, TIM3, LAG3 or CD160 were assessed by flow cytometry. Flow cytometry Zebra plots for Tbet+EOMES+ CD8+ T cells expressing PD-1 (N = 12–15). Statistical significance was tested using repeated measures of ANOVA followed by Tukey's posttest. *p<0.05, **p<0.01, ***p<0.001. Data are shown as mean ± SEM.

The CD8+Tbet+ cells with PD-1 expression (mean 38% vs 20.8, p=0.002: mean 38% vs 26.8, p=0.018) were decreased upon exposure to C-HIV and CI-HIV as compared to the levels found in F-HIV exposed cells (*Figure 6C*). The representative viSNE plot indicates visibly lower PD-1+Tbet+ clusters on C-HIV treated CD8+ T cells compared to F-HIV alone, while the levels were comparable when Mock and F-HIV were considered (*Figure 6C*). HIV exposure resulted in a similar profile in CD160+Tbet+CD8+ T cells (mean 61% vs 47.2, p=0.037: mean 61% vs 21.2, p=0.046) as for PD-1 with significant changes for the cells exposed to C-HIV and CI-HIV when compared to unexposed cells (*Figure 6C*). The levels of CD8+Tbet+ T cells expressing LAG3 or TIM3 remained similar across all the experimental conditions (*Figure 6C*). CD8+ T cells with co-expression of Tbet and EOMES were decreased in the isolated immune cells exposed to C-HIV (*Figure 6D*). Negative immune checkpoint molecules LAG3, TIM3 or CD160 expressed by Tbet+EOMES+CD8+ T cells were not altered by HIV exposure, whereas the levels of CD8+ T cells with PD-1 expression was significantly lowered following exposure to C-HIV and CI-HIV as compared to unexposed cells (*Figure 6D*). In summary, exposure to opsonized HIV decreased the CD8+ T cell expression of the negative checkpoint PD-1 and exposure to F-HIV increased the CD4 T cell expression of PD-1 and LAG-3. In addition, the effects on the on the T cell expression of transcription factors were similar between the C-HIV and CI-HIV exposed cells.

## HIV exposure alters the expression of perforin and granzyme in mucosal colorectal immune cells

CD4+ cytotoxic and CD8+ cytotoxic lymphocytes are known to directly control HIV replication and check viremia via secretion of lytic proteins, especially perforin and granzyme (*Kuerten et al., 2008*; *Hersperger et al., 2010*; *Juno et al., 2017*). The level of perforin and granzyme positive CD8+ T cells have been assessed before and the perforin levels in colon tissue are lower than in blood (*Kiniry et al., 2017*; *Shacklett et al., 2004*; *Shacklett et al., 2018*). In addition, transcription factors such as Tbet and EOMES are also involved in CD4+ and CD8+ T cell cytolytic functions (*Hersperger et al., 2010*; *Soghoian and Streeck, 2010*). Hence, we set out to analyze the expression of perforin and granzyme in CD4+ and CD8+ T cells isolated from colorectal tissue. Perforin expressing CD4+ T cells were significantly increased in the presence of F-HIV (mean 5.4% vs 10.9%, p=0.0004), C-HIV (mean 5.4% vs 9.6%, p=0.0009) and CI-HIV (mean 5.4% vs 11.3%, p=0.0002) (*Figure 7A*), while significantly higher levels of granzyme expression was recorded only for F-HIV-exposed CD4+ T cells (mean 4.9% vs 9.4%, p=0.031) (*Figure 7A*). The frequency of perforin positive CD8+ T cells was significantly lower in C-HIV (mean 11% vs 6.4%, p=0.0021) and CI-HIV (mean 11% vs 6.3%, p=0.0012) exposed groups as compared to F-HIV-exposed cells (*Figure 7B*), whereas there was only a significant decline in granzyme positive CI-HIV exposed CD8+ T cells as compared to unexposed cells (*Figure 7B*). In summary, complement opsonization of HIV is associated with impaired cytolytic capacity of CD8+ T cells, whereas the cytolytic activity of CD4+ T cells tends to increase following any HIV exposure.

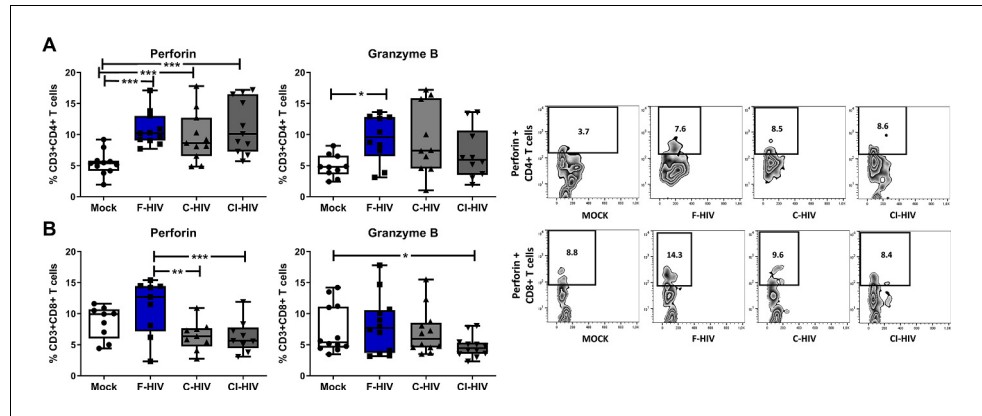

**Figure 7.** HIV exposure alters the expression of perforin and granzyme in mucosal cells. Mucosal immune cells isolated from the colorectal tissue biopsies by enzymatic digestion were exposed to HIV-1$_{BaL}$ (250 ng/mL), either free (F-HIV), complement-opsonized (C-HIV), or virions opsonized by a cocktail of complement and antibodies (CI-HIV) or mock-treated, by spinning the cultures. The effect HIV exposure had on mucosal T cells phenotype was assessed after 4 days. (**A and B**) The immune cells were stained with CD3, CD4, CD8, perforin and granzyme B mAbs. Of unexposed, F-HIV, C-HIV, or CI-HIV exposed CD4+ T cells (**A**) or CD8 T cells (**B**) expressing perforin or granzyme B were assessed by flow cytometry and shown as Zebra blots representing one individual and graphs for a summary of donors assessed. (**A**) Percentage of unexposed, F-HIV, C-HIV, or CI-HIV exposed CD4+ T cells expressing perforin or granzyme B were assessed by flow cytometry. (**B**) Percentage of unexposed, F-HIV, C-HIV, or CI-HIV exposed CD8+ T cells expressing perforin or granzyme B were assessed by flow cytometry (N = 12–15). Statistical significance was tested using repeated measures of ANOVA followed by Tukey's posttest. *p<0.05, **p<0.01, ***p<0.001. Data are shown as mean ± SEM.

## Complement opsonized HIV exposure led to a persistent low degree of inflammation in colorectal mucosal immune cells

The hallmark of HIV infection is the persistent production of pro-inflammatory cytokines and chemo-kines (*Ellegård et al., 2015*; *Freeman et al., 2016*; *Devalraju et al., 2018*). The gut mucosa in HIV-infected individuals despite the initiation of ART maintains a low level of inflammation, and the immune cells fail to recover completely (*McGowan et al., 2004*). Next, we analyzed the protein levels of IL-1β, IL-6, IL-17A, TNF, IL-10, CCL3 and CXCL10 at an early (24 hr) and later (96 hr) time-points, i.e. correlating to the time points for the transcriptome analysis of the colorectal tissue and the assessed of the effect on the isolated mucosal T cells. At 24 hr, the levels of IL-17A were signifi-cantly increased in cells following exposure to F-HIV (mean 32.7% vs 55%, p=0.011) and CI-HIV (mean 32.7% vs 62.6%, p=0.009) (*Figure 8A*) but this was lost at the later timepoint i.e. 96 hr. For the other proteins, there were a major range in the initial levels expressed and so it was difficult to establish any clear pattern related to different HIV conditions. At 96 hr, we found high levels of inflammatory factors in the C-HIV exposed mucosal immune cells, with significantly higher levels of IL-1β (mean 1.8 pg/ml vs 3.7 pg/ml, p=0.0039: mean 1.8 pg/ml vs 4.7 pg/ml, p=0.0018), IL-6 (mean 2485 pg/ml vs 4179 pg/ml, p=0.018: mean 2485 pg/ml vs 4207 pg/ml, p=0.015) and CXCL10 (mean 2.2 pg/ml vs 11 pg/ml, p=0.036: mean 2.2 pg/ml vs 14.1 pg/ml, p=0.0028) for C-HIV and CI-HIV exposed cells, and CCL3 for CI-HIV exposed cells as compared to unexposed cells (*Figure 8B*). C-HIV and CI-HIV induced persistent signs of inflammation at the 96 hr timepoint, a finding that is in line with decreased activation threshold of CD8+ T cells in combinations with a decreased ability to kill infected cells.

## Discussion

The genital and rectal mucosa are the initial sites of HIV dissemination, and the virus is more easily transmitted rectally due to the presence of a single layer of epithelium and high numbers of lym-phoid cells, such as DCs, macrophages and T cells, which are highly susceptible to HIV infection. The presence of these activated immune cells is the reason why the GI tract is the area hit hardest by HIV

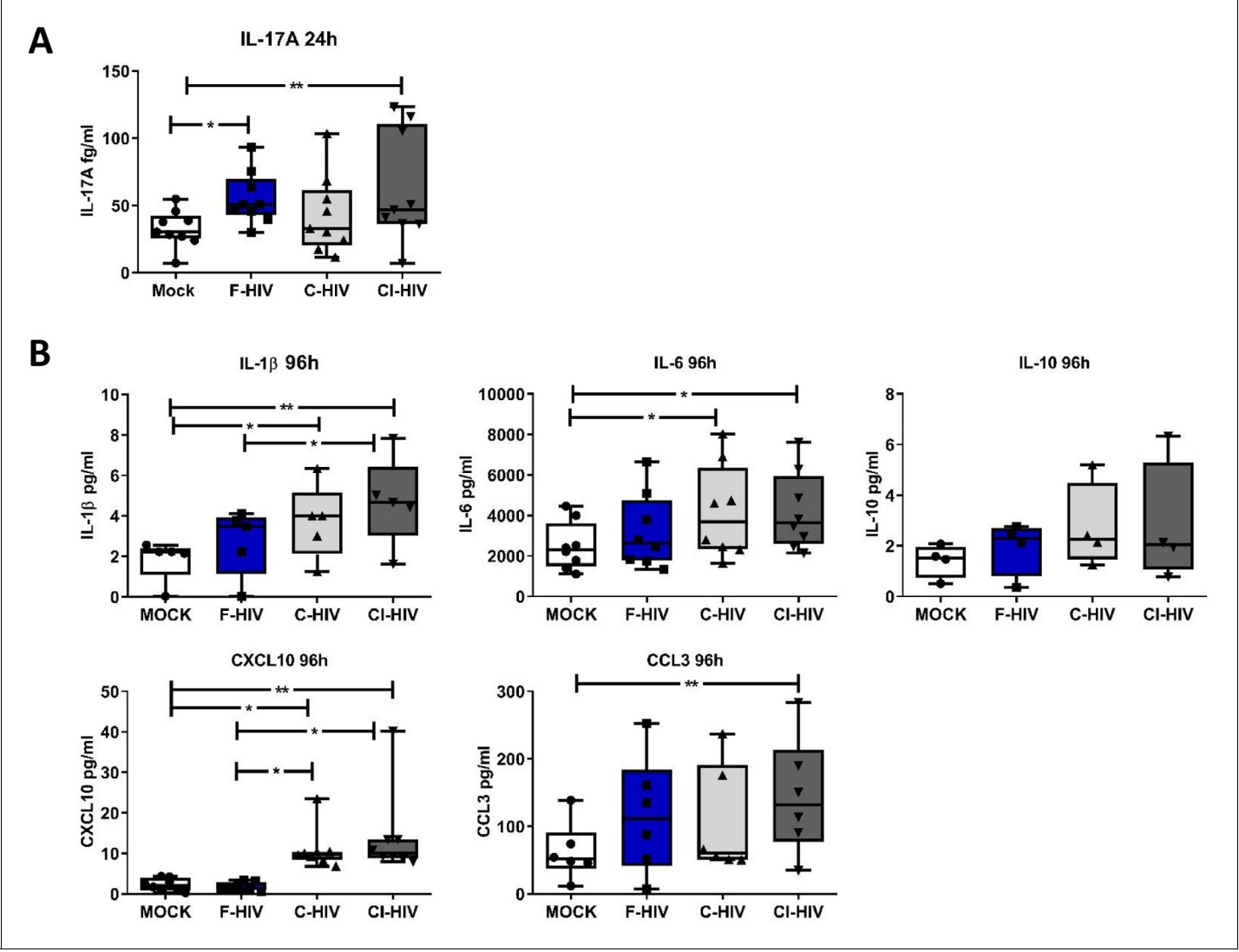

**Figure 8.** Complement-opsonized HIV gave rise to a delayed inflammation in the mucosal immune cells. Mucosal immune cells isolated from the colorectal tissue biopsies by enzymatic digestion were exposed to HIV-1$_{BaL}$ (250 ng/mL), either free (F-HIV), complement-opsonized (C-HIV), or virions opsonized by a cocktail of complement and antibodies (CI-HIV) or mock-treated, by spinning the cultures. The effect HIV exposure had on mucosal T cells phenotype was assessed after 24 h-4 days. (A) The supernatants from unexposed and F-HIV, C-HIV, or CI-HIV-exposed mucosal immune cells were assessed at 24 hr for IL-17A by bead array. (B) The supernatants from unexposed and F-HIV, C-HIV, or CI-HIV-exposed mucosal immune cells were assessed at 4 days for IL-6, IL-1β, IL-6, IL-10, CXCL10 and CCL3 by CBA (N = 12–15). Statistical significance was tested using repeated measures of ANOVA followed by Tukey's posttest. *p<0.05, **p<0.01, ***p<0.001. Data a shown as mean ± SEM.

(**Brenchley and Douek, 2008**). The initial events of HIV infection involving the human mucosa are still not well deciphered. In this study we aimed to elucidate the early events of HIV infection associated with the colorectal mucosa, using an explant model of human colorectal tissue and isolated immune cells from human colorectal mucosa. Tissues and cells were exposed to free and complement opsonized HIV. The initial responses induced by free HIV in the mucosal tissue were potent type I IFN and antiviral responses that were suppressed when the HIV was complement opsonized. Exposure to HIV modulated the immune transcriptome profiles at 24 hr in the mucosa to more activated T cell- and Th17-driven responses, which were in part mirrored in the mucosal T cell responses seen 96 hr after HIV exposure in the isolated mucosal immune cells. The strongest innate responses were seen in tissue and isolated cells exposed to F-HIV. There was elevated HIV infection in isolated and emigrated mucosal DCs when the virions were opsonized with complement, whereas the

emigrated T cells in most individuals had similar levels of infection independent of the type of HIV exposure. Exposure to F-HIV, increased number of CD4+ T cells expressing certain negative immune checkpoint factors. The exposure to HIV and the microenvironment created transformed the CD4+ T-cells into more inflammatory populations, that is Th17 and Th1Th17 cells. Noteworthy, even in CD8+ T cells the conditioning effects exerted by HIV and HIV infected cell exposure, especially when complement-opsonized, altered their activation status and the levels of perforin expression.

The early-transmitting HIV-1 virions that succeed to establish a productive infection in vivo are CCR5 tropic and the ability to infect the α4β7 integrin+CD4+ CCR5+ T cell population in the mucosa (*Nawaz et al., 2011*; *Peachman et al., 2015*; *Joseph et al., 2015*). HIV-1 BaL, which is a widely used laboratory HIV-1 strain and common tool to study HIV infection in ex vivo mucosa models was used in this study (*Merbah et al., 2012*; *King et al., 2013*; *Shen et al., 2012*), and the findings show that there were little to no differences in HIV infection of CD4+ T cells between HIV-1 BaL and founder/transmitter HIV strains. According to Parrish et al, the CCR5 tropic transmitted/founder virions are efficiently captured by monocyte-derived DCs and the transfer to CD4+ T cells is higher than for virions derived from chronic HIV-1 infected individuals (*Parrish et al., 2013*). The level of HIV-1 infection in mucosal and skin immune cells varies between different studies (*Tjomsland et al., 2013a*; *Bertram et al., 2019*; *Hladik et al., 2007*; *Introini et al., 2014*; *Merbah et al., 2011*).

Complement plays an important role in both innate and adaptive immunity and is of importance for the pathogenesis in many chronic inflammatory diseases (*Reis et al., 2019*). There are complement factors present in the colorectal mucosa, with both a local production by e.g. epithelial cells and a secretion of complement factors into the duodenum (*Andoh et al., 1996*; *Jain et al., 2014*; *Lai A Fat et al., 1976*). HIV-1 virions, both from transmitted founder and chronic isolates, and propagated from human cells have complement inhibitory molecules, such as CD55 and CD59, incorporated into their lipid membrane, which stops the complement cascade and protect them from being lysed (*Stoiber et al., 2008*). This gives rise to HIV-1 particles opsonized by inactivated complement fragments such as iC3b (*Bajtay et al., 2004*; *Ellegård et al., 2014*; *Saifuddin et al., 1995*). The HIV-1 BaL laboratory strain used in this study activated the complement cascade as measured by iC3b production (*Schumacher and Schreiber, 2015*) and the virions were opsonized by iC3b. Besides the iC3b there should be C1q fixation and especially in the CI-HIV group which was opsonized by serum and HIV-1 specific antibodies. Hence, the C-HIV and CI-HIV should reflect the virions transferred during sexual transmission. We have previously investigated HIV infection of the cervical mucosal tissue, and found that complement opsonization elevated HIV infection of emigrating DCs but not in the emigrating CD4+ T cells (*Tjomsland et al., 2013a*). The effects the opsonization of HIV exert on colorectal mucosal tissue or the mucosal DCs, macrophages and CD4+ T cells responses have, as far as we are aware, not been shown previously. The effects complement opsonization exerted on the infection and colorectal mucosal immune cells by the exposure to complement opsonized HIV should be similar in other mucosa tissues seeing the existence of complement components and regulators in tissues/body fluids, but the immunological/cellular effects on individual immune cell subsets ought to reflect the activation status of the tissue resident cells existing at different anatomical sites. Mucosal explant models are important tools to study the primary events of HIV-1 infection and can be used for testing new anti-viral drugs. A limitation of ex vivo tissue models is that they do not allow investigation of the influence of HIV on immune cells recruited to the site of infection from circulations.

Our previous investigation of HIV infection involving the cervical mucosal tissue, illustrated that complement opsonization elevated HIV infection of emigrating DCs but not in the emigrating CD4+ T cells (*Tjomsland et al., 2013a*).

Evidence suggests that complement opsonization of HIV (*Ellegård et al., 2015*; *Ellegård et al., 2014*; Svanberg et al., in preparation) and pathogens such as *Francisella tularensis* (*Dai et al., 2013*) has the ability to initially suppress antiviral and inflammatory responses when targeting complement receptor three and in the case of HIV rewire the signaling cascade, conferring HIV the window to infect target cells, which could be an explanation for the elevated infection. Of note, not all studies of complement opsonization of pathogens find this suppression.

The memory differentiation status for CD4+ T cells and CD8+ T cells was not substantially affected by HIV exposure in our study, and in the colorectal tissue the CD45RA-CCR7- effector memory T cells remained the dominating T cell phenotype. The levels of the more terminally differentiated effector memory T cells CD45RA+CCR7- population was higher in the CD8+ T cell population

than in the CD4+ T cell population, which is in line with findings from peripheral blood. Noteworthy, the conditioning by HIV, especially in the F-HIV and CI-groups enhanced the frequency of CD4+ T cells expressing CXCR3+CCR6+. This cell type in blood has been shown to be highly susceptible to HIV-1 infection and to have gut homing abilities (*Gosselin et al., 2010*). Furthermore, CXCR3+CCR6 + CD4+ T cells are one of cell types that is decreased in HIV-1 infected individuals even when on ART (*Gosselin et al., 2010*). In chronic SIV infection, there is an increase in the level of blood CXCR3 + CD4+ T cells, this is also reflected in the lymph nodes where CXCR3+ T follicular helper cells (Tfh) are known to harbor high levels of virions (*Velu et al., 2016*).

Tbet was originally considered as an essential Th1 CD4+ T cell regulating factor with the ability to impair both Th2 and Th17 development, and to maintain memory CD4+ and CD8+ T-cell subsets (*Pipkin et al., 2010*). Additionally, Tbet has the ability to regulate several transcription networks such as T cell migration and cytolytic signaling molecules (*Lazarevic and Glimcher, 2011*) and high levels of Tbet have been shown to correlate with CD8+ T cell upregulation of perforin and granzyme B (*Hersperger et al., 2010*). Our investigations found alteration of the cytotoxic CD4+ and CD8+ T cell populations in the isolated mucosal immune cells after HIV exposure. The levels of CD4+ T cells with perforin and/or granzyme B expression increased, whereas the amount of perforin+ CD8+ T cells decreased. The observation of low levels of CD8+ T cells expressing perforin after HIV exposure is clearly in agreement with our previous data where the NK cells ability to kill target cells was decreased when activated by DCs exposed to C-HIV. In addition, the level of perforin in T cells primed by C-HIV and CI-HIV exposed DC-NK cell cocultures was low (*Ellegård et al., 2018*). Furthermore, this decrease of perforin-expressing CD8+ T cells could be linked to the decreased levels of Tbet and/or EOMES positive cells indicating that the cytotoxic functionality of CD8+ T cells is regulated by these transcription factors (*Cruz-Guilloty et al., 2009*). If these findings truly reflect the in vivo circumstances in the gut during the onset of HIV infection, these activated CD8+ T cells with decreased killing abilities would be inadequate to control the infection.

T cell suppression, marked by loss of effector functions and increased expression of different coinhibitory/negative checkpoint molecules, is common in chronic viral infections like HIV (*Wherry and Kurachi, 2015*). Our study showed that during the initial phases of HIV exposure an increase in the expression of negative immune checkpoint molecules was noticeable on CD4+ T cells, especially after F-HIV exposure, indicating that exposure to HIV and infection of CD4+ T cells leads to cells with higher activation threshold together with potentially suppressive abilities. The CD8+ T cell populations with negative immune checkpoint factors did not increase, instead in the case of PD-1 and LAG3 a decrease was seen. Complement opsonized HIV reduced the levels of colorectal CD8+ T cells expressing PD-1, which points to impaired ability to regulation of immune responses. A recent study in individuals with psoriasis found decreased PD-1 expression on CD4+ and CD8+ T cells, which suggested that this decrease could likely play a key role in chronic immune activation (*Bartosińska et al., 2017*).

In untreated HIV infection, CD38 expression by blood T cells is an indication of T cell impairment and HIV disease progression (*Liu et al., 2017*). Under normal circumstances, CD38 is highly expressed on activated T cells in cervical mucosal explants, and low frequency of CD38+CD4+ and CD8+ T cells was associated with reduced HIV replication (*Saba et al., 2017*) indicating that CD38 expression on mucosal T cells either supports or is a product of HIV infection. Furthermore, in HIV-infected individuals failing to control infection, the co-expression of CD38 and PD-1 on CD4+ and CD8+ T cells could be a sign of T-cell activation (*Shaw et al., 2011*). The frequency of CD8+ T cells isolated from colorectal tissue that expressed CD38 was not affected by the HIV exposure but the levels of CD38+CD8+ T cells expressing PD-1 and CD160 decreased, which should likely lower their activation threshold. A recent study examining HIV infection of cervical mucosal explants established that low number of CD38+CD4+ and CD8+ T cells was associated with reduced HIV replication (*Saba et al., 2017*) indicating that CD38 expression on mucosal T cells likely favors productive HIV infection. Nonetheless, we did not assess the frequency of mucosal CD38+CD4+ T cells in this study, and hence we were unable to conclude if this was the case in our study. In our previous study, we found increased levels of CD38+CD4+ T cells indicating that CD4+ T cells primed by DCs exposed to C-HIV could augment the rates of HIV replication (*Ellegård et al., 2018*).

There was a clear indication from the transcriptome analysis of mucosal tissue and the phenotypic assessment of mucosal immune cells that the Th17 pathway was activated following HIV exposure. In particular, F-HIV-activated factors are involved in the differentiation and maintenance of Th17

responses at the transcription level especially for IL-16, IL-17A, IL-23, IFN-γ and IL-1β (*Fernandes et al., 2017*) and at the protein level with IL-17A. This activation was supported by phenotypic analysis, which clearly showed an increase in Th17 population among HIV-exposed cells, that is CD4+ T cells expressing RORγt and, CCR6+, as well as increased production of IL-17A.

In this study, we have shown that primary HIV infection of colorectal mucosa in vitro lead to initial strong antiviral responses when induced by free virus that in part protected and created an environment with a lower level of infection as compared to complement opsonized HIV. Opsonization altered the transcriptome profiles, with the free HIV activating different innate and inflammatory pathways during early stages of infection compared to complement opsonized HIV. It is also clear that the environment created by complement opsonized HIV with an initial suppressed antiviral response in the mucosal tissue and immune cells contribute to the higher levels of HIV-infected CD4 + T cells and later an environment with higher inflammatory responses. Furthermore, there were increased frequencies of CD4+ T cells with PD-1 expression in association with Tbet/EOMES expression when exposed to free HIV, compared to complement opsonized HIV. In the CD8+ T cells, the PD-1 expression dropped after exposure to opsonized HIV compared to free HIV in Tbet/EOMES+ T cells. Complement opsonized HIV exposure decreased the levels of perforin expressing CD8+ T cells, whereas the expression increased on CD4+ T cells after both free and complement opsonized HIV exposure. In conclusion, opsonization of HIV alters the activation of signaling pathways and cells in the colorectal mucosa in a manner that promotes viral establishment and by creating an environment that stimulates mucosal T cell activation and inflammatory T helper cells, and this could play an important role in HIV immunopathogenesis.

# Materials and methods

**Key resources table**

| Reagent type (species) or resource | Designation | Source or reference | Identifiers | Additional information |
|---|---|---|---|---|
| Biological sample (human) | Colorectal Tissue Biopsies | Surgical Department of Linköping University Hospital | | The tissue pieces were either cut into small explants or punched into biopsies |
| Antibody | anti-human CD3-PerCP/AmCyan(Mouse monoclonal) | BD Biosciences | Cat#: 552851/ 339186 | FACS (1 ul per test) |
| | anti-human CD4-PE/BV421(Mouse monoclonal) | BD Biosciences | Cat#: 555347/ 562424 | FACS (1 ul per test) |
| | anti-human CD8-PerCPCy5.5(Mouse monoclonal) | BD Biosciences | Cat#: 560662 | FACS (1 ul per test) |
| | anti-human CD1a-APC(Mouse monoclonal) | BD Biosciences | Cat#: 559775 | FACS (1 ul per test) |
| | anti-human CD163-APCCy7(Mouse monoclonal) | BioLegend | Cat#: 50-850-648 | FACS (1 ul per test) |
| | anti-human CD45RA-FITC(Mouse monoclonal) | BD Biosciences | Cat#: 555488 | FACS (1 ul per test) |
| | anti-human CD45RO-FITC(Mouse monoclonal) | BD Biosciences | Cat#: 555492 | FACS (1 ul per test) |
| | anti-human CD38-PerCP-Cy5.5(Mouse monoclonal) | BD Biosciences | Cat#: 551400 | FACS (1 ul per test) |
| | anti-human CD160-PE(Mouse monoclonal) | BD Biosciences | Cat#: 562118 | FACS (1 ul per test) |
| | anti-human PD-1-BV421(Mouse monoclonal) | BioLegend | Cat#: 329920 | FACS (1 ul per test) |
| | anti-human TIM3-PE(Mouse monoclonal) | BioLegend | Cat#: 345006 | FACS (1 ul per test) |

*Continued on next page*

*Continued*

| Reagent type (species) or resource | Designation | Source or reference | Identifiers | Additional information |
|---|---|---|---|---|
| | anti-human LAG3-FITC(Mouse monoclonal) | BioLegend | Cat#: 369308 | FACS (1 ul per test) |
| | anti-human CXCR3-PE(Mouse monoclonal) | BioLegend | Cat#: 353706 | FACS (1 ul per test) |
| | anti-human CCR4-AF647(Mouse monoclonal) | BioLegend | Cat#: 359404 | FACS (1 ul per test) |
| | anti-human CCR6-PECy7(Mouse monoclonal) | BD Biosciences | Cat#: 560620 | FACS (1 ul per test) |
| | anti-human CCR7-PerCPCy5.5(Mouse monoclonal) | BioLegend | Cat#: 353220 | FACS (1 ul per test) |
| | anti-Tbet-FITC/PECy7(Mouse monoclonal) | BioLegend | Cat#: 644812/ 644824 | FACS (1 ul per test) |
| | anti-human EOMES-eFluor660(Mouse monoclonal) | eBioscience | Cat#: 50-4877-41 | FACS (1 ul per test) |
| | anti-human RORγt-PE(Mouse monoclonal) | BD Biosciences | Cat#: 563081 | FACS (1 ul per test) |
| | anti-GATA3-PECy7(Mouse monoclonal) | BD Biosciences | Cat#: 560405 | FACS (1 ul per test) |
| | anti-human FoxP3-AF647(Mouse monoclonal) | BD Biosciences | Cat#: 560045 | FACS (1 ul per test) |
| | anti-human Perforin-Pacific Blue(Mouse monoclonal) | BioLegend | Cat#: 308118 | FACS (1 ul per test) |
| | anti-human Granzyme B-AF647(Mouse monoclonal) | BD Biosciences | Cat#: 560212 | FACS (1 ul per test) |
| | Anti-human HIV-1 Core Antigen-FITC, KC57 (Mouse monoclonal) | Beckman Coulter | Cat#: 6604665 | FACS (1 ul per test) |
| | Anti-human IgG1 Mouse-FITC Isotype Control | Beckman Coulter | Cat#: IM0639U | FACS (1 ul per test) |
| | Zombie NIR Fixable Viability Kit | BioLegend | Cat#: 423106 | FACS (diluted as manufacturer's protocol) |
| Sequence-based reagent | β-Actin_F | Eurofins Genomics/ CyberGene AB | PCR primers | |
| | β-Actin_R | Eurofins Genomics/ CyberGene AB | PCR primers | |
| | GADPH_F | Eurofins Genomics/ CyberGene AB | PCR primers | |
| | GADPH_R | Eurofins Genomics/ CyberGene AB | PCR primers | |
| Commercial assay or kit | Human Inflammatory Cytokine CBA Kit | BD Biosciences | 551811 | |
| | Human IFNγ CBA Flex Set | BD Biosciences | 558269 | |
| | Human IP-10 CBA Flex Set | BD Biosciences | 558280 | |
| | Human MIP-1α CBA Flex Set | BD Biosciences | 558325 | |
| | Human IL-17A Enhanced Sensitivity CBA Flex Set | BD Biosciences | 562143 | |
| Software, algorithm | CFX Manager 3.1 | Bio-Rad | | |
| | Ingenuity Pathway Analysis (IPA) | Qiagen | | |
| | FlowJo | Treestar | | |
| | FCAP Array | BD Biosciences | | |
| | GraphPad Prism 5 | GraphPad Software | | |

## Ethics statement

This study was approved by the Linköping University Ethical Review Board (Ethical permit EPN M206-06). The subjects were informed about the study at the clinic and verbal consents were obtained and documented from all participating subjects, as approved by the Linköping University Ethical Review Board. The study included both male and female adult subjects who were 18 years or older. Normal colorectal tissue samples (N = 45) were obtained from individuals undergoing surgery for colorectal cancer at the Surgical Clinic of Linköping and Norrköping Hospitals, Sweden. The tissue biopsies obtained from surgery were subjected to histopathological examination by the clinical pathologist and showed no histological signs of malignancy, infection or inflammation (*Figure 1— figure supplement 1*).

## Virus generation and opsonization

HIV-1BaL ((lot no. 4238) Biological Products Core/AIDS and Cancer Virus Program, SAIC-Frederick, Inc, NCI Frederick) was produced from SUP-T1/CCR5 cells, derived from T cell lymphoblastic lymphoma, and purified as described previously (*Rossio et al., 1998*). Opsonization was carried out as described previously (*Bajtay et al., 2004*; *Ellegård et al., 2014*; *Tjomsland et al., 2013b*), in brief the virus (250 ng/ml) was incubated for 1 hr at 37°C with either an equal volume of RPMI1640 (Sigma-Aldrich, Stockholm, Sweden) to generate free HIV (F-HIV), single-donor human serum to generate complement-opsonized HIV (C-HIV), or single donor human serum supplemented with 0.02 µg/ml HIV-specific IgG (SMI, Stockholm, Sweden) and 20 µg/ml γ-globulins (Pharmacia, Stockholm, Sweden) to generate complement- and antibody-opsonized HIV (CI-HIV). The single-donor human serum samples were collected from several donors throughout the study. The ability to opsonize the virions was assessed with functional assays such as enhanced infection of DCs (*Bajtay et al., 2004*; *Bouhlal et al., 2002*; *Bouhlal et al., 2007*; *Posch et al., 2015*) and iC3b production (*Figure 1—figure supplement 2A–C*) as a measurement of complement activation induced by HIV during opsonization in serum (*Schumacher and Schreiber, 2015*) and levels of iC3b bound to the virions (*Figure 1—figure supplement 2A–C*) using an iC3b ELISA (2BScientific Limited, Oxfordshire, United Kingdom). The binding of IgG to the virions was assessed using an IgG ELISA (MABTECH, Stockholm, Sweden) (*Figure 1—figure supplement 2A–C*). In brief, to measure the iC3b and IgG bound to the virus, the virions were left untreated or opsonized as described above followed by purification by centrifugation in an ultracentrifuge (Beckman) using a SW55Ti rotor for 2 hr at 32500 RPM. The viral pellets were resuspended in PBS and then lysed, and the level of iC3b and IgG measured by ELISA.

## Processing of colorectal tissue, infection, and cell culture

Colorectal tissue biopsies, as obtained from the Surgical Department of Linköping University Hospital, were transported in aerated (with a gas containing 95% $O_2$ and 5% $CO_2$ for 10 min) Krebs buffer on ice. The isolation of cells started within 1–2 hr from collection. The biopsy specimen was first cleaned with RPMI-1640, cut open longitudinally, to reach to the soft mucosal layer of tissue. The rigid muscular layer together with fat were then removed with the help of a pair of scissors and forceps. The cleaned mucosal tissues were either cut into small explants (8 mm$^2$) or punch biopsied (3 or 6 mm$^2$, PFM Medical, Solingen, Germany). Tissue samples were challenged with HIV-1BaL (250 ng/mL), spinocculated at 1200 RPM for 2 hr at 37°C, washed four times, transferred to six-well tissue culture plates, and cultured. The tissue biopsies for the same culture conditions was pooled before RNA extraction/analysis. For some experiments, the emigrating colorectal cells were collected after 3–4 days, stained for flow cytometry and fixed in 4% paraformaldehyde (PFA). In most experiments isolated mucosal immune cells were used. To isolate immune cells from the colorectal tissues, biopsies were rigorously minced and made into a slurry of cells, which was subsequently treated with 0.5 mg/ml collagenase type II (Sigma-Aldrich) and 0.05 mg/ml DNase (Roche Diagnostics, IN, USA) before incubation for 1 hr at 37°C with intermittent shaking. Tissues were further disrupted by forcing the suspension through a 20-cc syringe (BD Plastipaks, Stockholm, Sweden) with a blunt-ended needle (Stemcell Technologies, Cambridge, UK). Subsequently, the suspension was filtered by a metallic strainer (Sigma-Aldrich). Colorectal immune cells were isolated from the cell suspension by Ficoll density gradient centrifugation at 2200 g for 22 mins. The final number of mononuclear lymphocytes obtained depended on the size of the tissue, and in our setup varied from $5 \times 10^6$ to $2 \times 10^7$ cells. The cells ($5 \times 10^5$/well) were plated in 24-well cell culture plates (Corning, NY, USA) in RPMI-1640 (Sigma-Aldrich) with 5% PHS (Innovative Research, MI, USA), 20 µg/ml Gentamicin, and 10 mM HEPES (Fisher Scientific, Gothenburg, Sweden) and unexposed or exposed to F-HIV, C-HIV or CI-HIV (250 ng/mL). The colorectal immune cells were cultured for 24–96 hr.

## RNA extraction and quantitative real-time PCR

The isolated mucosal immune cells separated from colorectal immune cells were cultured with the different HIV groups for 24 hr and thereafter subjected to RNA extraction using a commercial Isolate II RNA Mini Kit (Bioline, Luckenwalde, Germany). The number of mucosal immune cells used for RNA extraction were between $5 \times 10^5$ to $1 \times 10^6$ in our setup. cDNA was produced using SuperScript III reverse transcriptase first-strand cDNA synthesis kit (Invitrogen, Carlsbad, CA). Quantitative real-

time PCR was performed with Master Mix (SensiFAST SYBR Hi-ROX Kit, Bioline) using a CFX96 Real-time PCR System (C100 Touch) with CFX Manager 3.1 software (Bio-Rad, CA, USA). Primers targeting β-actin and GADPH were used as housekeeping genes for reference as described by *Vandesompele et al., 2002*. Primers were purchased from Eurofins Genomics (Ebersberg, Germany) and CyberGene AB (Solna, Sweden). To compensate for variation between plates, the values were normalized as described by *Rieu and Powers, 2009*.

## RNA sequencing and data analysis

Whole transcriptome amplification of RNA purified from colon explants cultured with different forms of HIV (F-HIV, C-FIV and CI-HIV) or untreated (mock) for 24 hr was performed using a commercial NuGEN's Ovation RNA-Seq V2 kit following the manufacturers protocol (San Carlos, CA, USA). cDNA was amplified from total RNA using a single primer isothermal amplification and purified using a MinElute Reaction Cleanup Kit (Qiagen, CA, USA). The cDNA samples were fragmented, barcoded with adaptors, and amplified using an Ultralow System V2 kit. Distribution of the size of the library was determined using an Agilent Bioanalyzer 2100. Libraries from three different donors were sequenced on the Illumina NextSeq500 platform (San Diego, CA, USA). The fastq files were uploaded and the quality checked using the fastQC program (*Babraham-bioinformatics, 2010*). Trimmomatic (*Bolger et al., 2014*) was used to remove adaptors and low-quality bases, and the reads were then mapped to the human reference genome hg19 using STAR. FeatureCounts was used to calculate the counts for each gene (*Liao et al., 2014*). The data were thereafter normalized and R/DeSeq2 was used to determine differentially expressed genes (*Love et al., 2014*). Analysis of pathways was done by Ingenuity Pathway Analysis (Qiagen) to assess canonical pathways, disease and biofunctions as well as up-stream regulators that was either up- or down-regulated using Z-scores calculated by the program, R analysis, and custom gene lists. A p-value cut-off of 0.05 was set as significant for affected molecules/pathways.

## Flow cytometry

Isolated mucosal immune cells were harvested after 3 or 4 days of culture, washed and suspended in FACS buffer (0.2% FCS in PBS) and stained with antibodies specific for CD3, CD4, CD8, CD1a, CD163, CD45RA, CD45RO, CD38, CD160, PD-1 (CD279), TIM3 (CD366), LAG3 (CD223), CXCR3 (CD183), CCR4 (CD194), CCR6 (CD196) and CCR7 (CD197) (BD Biosciences and BioLegend (CA, USA)) for 30 mins. The cells were fixed in 4% PFA, permeabilized in 0.2% saponin (Sigma-Aldrich) before intracellular staining with antibodies specific for Tbet, EOMES, RORγt, GATA3, FoxP3, perforin and granzyme B (BD Biosciences and BioLegend) for 45 mins. HIV-1 p24 was detected with anti-HIV-1 mAb (KC57, clone FH190-1-1, Beckman Coulter) and the corresponding isotype control (Beckman Coulter). Cell acquisition was performed using FACS Diva (BD Biosciences) software on a FACSCanto II flow cytometer (BD Biosciences) and data were analyzed by FlowJo (Treestar, OR, USA). Cell viability was assessed by Zombie NIR Fixable Viability Kit (BioLegend), and to further exclude any dead cells/debris, doublet discrimination through forward scatter-area/forward scatter-width gating was performed along with normal forward scatter-area/side scatter-area gating. The gating strategy for live lymphocytes is available in supplementary information (*Figure 2—figure supplement 1*). This population was then subjected to viSNE analysis to obtain a two-dimensional (tSNE1/tSNE2) plot exhibiting distribution of total lymphocytes. Parameters considered to construct the 2D plots include lymphocytes, CD4 or CD8 T cells and Tbet+PD-1+ depending on experiments. The heat gradient (blue to red) indicates expression levels of subsets as mentioned, following exposure to different conditions of viruses, with higher red dots implying higher percentages.

## Cytometric bead array (CBA)

The protein levels of IFN-γ, IL-10, IL-6, CXCL8, IL-1β, TNF, CXCL10 and CCL3 were measured in supernatants from virus-infected colorectal cells both at 24 and 96 hr using CBA Flex Sets (BD Biosciences, Stockholm, Sweden). For assessment of IL-17A levels, Enhanced Sensitivity Flex Set (BD Biosciences) was used. Samples were processed and run in a BD FACSCanto II flow cytometer (BD Biosciences) and analyzed using a FCAP Array version three software (BD Biosciences) according to the manufacturer's protocol.

## Statistical analysis

The data obtained by RNA sequencing of colorectal tissue was normalized and R/DeSeq2 used to determine differentially expressed genes (*Love et al., 2014*). Analyses of pathways were done by Ingenuity Pathway Analysis (Qiagen), R analysis, custom gene lists, and by GO analysis. All other results were analyzed using GraphPad Prism 5 (GraphPad Software, CA, USA), with either two-sided paired t-test or repeated measures of ANOVA followed by Tukey's posttest. p<0.05 was considered statistically significant.

## Acknowledgements

This work has been supported by: AI52731, The Swedish Research Council, The Swedish Physicians against AIDS Research Foundation, The Swedish International Development Cooperation Agency, SIDA SARC, VINNMER for Vinnova, Linköping University Hospital Research Fund, CALF, and The Swedish Society of Medicine for ML. The computations for the RNAseq were performed on resources provided by SNIC through Uppsala Multidisciplinary Center for Advanced Computational Science (UPPMAX) under project b2015293.

## Additional information

### Funding

| Funder | Grant reference number | Author |
| --- | --- | --- |
| Vetenskapsrådet | Project grant AI52731 | Marie Larsson |
| Stiftelsen Läkare mot AIDS Forskningsfond | Project grant | Marie Larsson |
| Forsknings-ALF | Project grant | Marie Larsson |
| Styrelsen för Internationellt Utvecklingssamarbete | | Marie Larsson |
| SIDA SARC | | Marie Larsson |
| VINNOVA | VINNMER | Marie Larsson |
| Linköping University Hospital Research Fund | | Marie Larsson |
| Svenska Läkaresällskapet | | Marie Larsson |

The funders had no role in study design, data collection and interpretation, or the decision to submit the work for publication.

### Author contributions

Pradyot Bhattacharya, Data curation, Investigation, Methodology, Writing - original draft, Writing - review and editing; Rada Ellegård, Data curation, Formal analysis, Investigation, Methodology; Mohammad Khalid, Formal analysis, Investigation; Cecilia Svanberg, Melissa Govender, Sofia Nyström, Investigation, Writing - review and editing; Åsa V Keita, Resources, Investigation, Methodology; Johan D Söderholm, Methodology, Writing - review and editing; Pär Myrelid, Resources, Methodology, Writing - review and editing; Esaki M Shankar, Formal analysis, Investigation, Writing - review and editing; Marie Larsson, Conceptualization, Resources, Supervision, Funding acquisition, Writing - original draft, Writing - review and editing

### Author ORCIDs

Marie Larsson (iD) https://orcid.org/0000-0002-4524-0177

### Ethics

Human subjects: This study was approved by the Linköping University Ethical Review Board (Ethical permit EPN M206-06). The subjects were informed about the study at the clinic and verbal consents were obtained and documented from all participating subjects, as approved by the Linköping

University Ethical Review Board. The study included both male and female adult subjects who were 18 years or older.

## Decision letter and Author response
Decision letter https://doi.org/10.7554/eLife.57869.sa1
Author response https://doi.org/10.7554/eLife.57869.sa2

# Additional files
## Supplementary files
• Transparent reporting form

## Data availability
Sequencing data (RNA seq) have been deposited in GEO, under the accession number GSE149749.

The following dataset was generated:

| Author(s) | Year | Dataset title | Dataset URL | Database and Identifier |
|---|---|---|---|---|
| Bhattacharya P, Ellegård R, Khalid M, Svanberg C, Govender M, Keita Å, Söderholm J, Myrelid P, Shankar E, Nyström S, Larsson M | 2020 | Colorectal mucosa exposed to free and complement opsonized HIV | https://www.ncbi.nlm.nih.gov/geo/query/acc.cgi?acc=GSE149749 | NCBI Gene Expression Omnibus, GSE149749 |

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
