## [Decision Letter]

**Acceptance summary:**

In this manuscript the authors have studied the early interaction of HIV-1 with cells within mucosal rectal tissue (surgical explants) to study the effects on induction of immune responses and consequences for HIV-1 infection. This work gives a comprehensive comparison of the differential regulation of transcriptional pathways, infection levels and cellular phenotypes in human colorectal tissue explants. They additionally monitor the effects between free HIV-1 particles in comparison to either opsonised virus (complement) or virus in the presence complement and antibodies. Their conclusions map a wide-range of variations in host cell induced responses pertaining to CD4 cell activation and in lowering of CD8 mediated cell responses. These results provide indications as to the early events in HIV-1 infection that can help explain for the rapid replication observed mucosal gut tissue. It is also interesting that complement, usually thought of as a key host defense pathway, could exacerbate infection and skew immune responses in a way that is not beneficial to the host.

**Decision letter after peer review:**

Thank you for submitting your article "Complement opsonization of HIV affects primary HIV infection of colorectal mucosa and subsequent activation of T cells" for consideration by *eLife*. Your article has been reviewed by three peer reviewers, including Nicola L Harris as the Reviewing Editor and Reviewer #1, and the evaluation has been overseen by Päivi Ojala as the Senior Editor. The following individuals involved in review of your submission have agreed to reveal their identity: William Paxton (Reviewer #2); Ashley L. St. John (Reviewer #3).

The reviewers have discussed the reviews with one another and the Reviewing Editor has drafted this decision to help you prepare a revised submission.

Summary:

In this manuscript the authors have studied the early interaction of HIV-1 with cells within mucosal rectal tissue (surgical explants) to study the effects on induction of immune responses and consequences for HIV-1 infection. They study the effects of HIV-1 on modulating CD4 and CD8 responses and measure alterations to a large array of host proteins involved with pro-inflammatory responses and immune induction. They additionally monitor the effects between free HIV-1 particles in comparison to either opsonised virus (complement) or virus in the presence complement and antibodies. Their conclusions map a wide-range of variations in host cell induced responses pertaining to CD4 cell activation and in lowering of CD8 mediated cell responses. These results provide indications as to the early events in HIV-1 infection that can help explain for the rapid replication observed mucosal gut tissue.

Essential revisions:

1) For all the experiments it would be important to determine whether the skewing is due to HIV induced tx factor and cytokine production or due to HIV induced death of sub populations of cells. For this the authors need to provide information on death of CD4 cell populations and whether this differs across conditions tested. If it does differ the specific death of skewed subsets needs to be determined as a possible factor contributing the findings.

2) Clarification regarding experimental details and statistical analyses are required. More detail in terms of data values should be given along with actual P values. E.g. where large increases/decreases are found state the fold changes in brackets and provide the p-value or where means are shown provide the value in brackets. Moreover, the statistical analysis of Figure 2D, F, I, H may not be correct because the fold change is set to 1 for all groups. This would lead to the variance for that group to be 0, which in an ANOVA could skew the results by suggesting lower variation than is real for one group. Although for one way to represent the data for clarity, but it would be better to perform statistics on raw data.

3) A control of inactivated virus with and without complement would be helpful to assess the contributions of signaling induced by the complement and/or complement/antibody complexes to the phenotype observed. Changes to infection levels under the various conditions were also described in the Results, but whether the CI-HIV and C-HIV groups resulted in changes to the viral tropism or infection of different proportions of cells was not reported but must be available from existing data.

4) Figure 4E-F the authors state that infection increases the expression of chemokine receptors, leading to more susceptible cells in the culture, but it would be important to show the chemokine receptor expression in the HIV+ versus HIV- populations for each tissue to clarify whether this is due to a bystander effect or if only those cells that are directly infected increase chemokine receptor expression, which could indicate an alternate interpretation for why the numbers of infected cells also increase.

---

## [Author Response]

Essential revisions:1) For all the experiments it would be important to determine whether the skewing is due to HIV induced tx factor and cytokine production or due to HIV induced death of sub populations of cells. For this the authors need to provide information on death of CD4 cell populations and whether this differs across conditions tested. If it does differ the specific death of skewed subsets needs to be determined as a possible factor contributing the findings.

We strongly believe that the initial HIV exposure program and condition the tissue/cells is responsible for the effects seen and should be a mix of receptor binding and subsequent signaling and secretion of factors that modulate the T cell phenotype/ response. When examining the live/death in the mock, F-HIV, C-HIV, and CI-HIV, we have found that percentages of live lymphocytes are more or less the same for all the experimental conditions, i.e. we do not see any significant differences between the mock and HIV groups. To confirm this, we have now examined the % live vs. dead T cells and we do not find any differences in the levels of live and dead cells among the T cell populations, which supports our interpretation that the effects seen are due to the initial HIV exposure. We have added the % live bulk CD3+ T cells to the Figure 3—figure supplement 1.

2) Clarification regarding experimental details and statistical analyses are required. More detail in terms of data values should be given along with actual P values. E.g. where large increases/decreases are found state the fold changes in brackets and provide the p-value or where means are shown provide the value in brackets. Moreover, the statistical analysis of Figure 2D, F, I, H may not be correct because the fold change is set to 1 for all groups. This would lead to the variance for that group to be 0, which in an ANOVA could skew the results by suggesting lower variation than is real for one group. Although for one way to represent the data for clarity, but it would be better to perform statistics on raw data.

We have now added the requested experimental details, i.e. mean values, fold change, and/or P values, in the Results text and/or figure legends. The reviewers are correct, ANOVA is the wrong statistics for Figure 2’s normalized data. We have now added the correct statistics to the Figure 2 legend (One- tailed Paired T-test).

3) A control of inactivated virus with and without complement would be helpful to assess the contributions of signaling induced by the complement and/or complement/antibody complexes to the phenotype observed. Changes to infection levels under the various conditions were also described in the Results, but whether the CI-HIV and C-HIV groups resulted in changes to the viral tropism or infection of different proportions of cells was not reported but must be available from existing data.

Inactivated virus: This is an excellent suggestion to investigate inactivated virus with and without complement and will be part of the follow up study focusing on single cell RNA seq and CyTOF/40 color flow. These methodologies will be even better tools to use to study the HIV effects in deeper details. We had planned to start with this during spring but had to postpone as we do not have access to colon tissue at the present time due to COVID-19. We have contacted the clinic providing the tissue and we will unfortunately not have access to tissue samples, and there is no indication of a time frame, which could range from 3 months to maybe a year before we are back to normal and can get tissues again.

We plan to use an inactivated, noninfectious virus, such as AT2 inactivated HIV-1, which can bind and fuse but not produce provirus or integrate with the genome. This would be a very interesting experiment. We hypothesize that free and complement opsonized AT2 HIV would give more or less the same transcriptome and PCR profiles as the live virions in the first 24 h. This is due to that we expect that a productive infection of HIV is not necessary to induce the effects seen at 24h. This initial conditioning by the complement opsonized HIV-1, seen at the transcriptomic level, gives rise to the effects seen later in the T cell populations and allows for higher level of infection. The effects that could/should be due to productive infection, i.e. later effects such as the inflammation at 96h, could be affected and altered if using an inactivated HIV-1. Other forms of inactivation, such as heat or PFA inactivation of the virions should not be good alternatives, seeing that these treatments have been shown to affect the envelope proteins structure and ability to bind to CD4 (Rossio et al., 1998), which could alter the complement opsonization of virions.

Tropism: The HIV-1 BaL is a CCR5 tropic virus and we have shown before that the CCR5 tropism remains when the virions are complement opsonized HIV in a previous study of cervical mucosa. We found that CCR5 blocking affected both the free and complement opsonized HIV infection of mucosal CD4^+^ T cells, with the highest inhibition in infection seen for the complement opsonized group (Tjomsland et al., 2013), which indicate that the opsonization do not shift from CCR5 usage and that the higher infection don’t depend on shift in co-receptor usage. This does not mean that the HIV-1 BaL not could take advantage of other receptors such as CXCR4 for infection. We believe that the infection is determined by many factors such as “help receptors” e.g. α4β7, DC-SIGN and in the case of complement opsonized receptors, e.g. DC-SIGN, FcRs and CR3, which together might help to facilitate the infection of CD4^+^ cells by a more efficient interaction and uptake of virions by the CD4^+^ T cells (Tjomsland EJI 2011, Tjomsland et al., 2013, Bouhlal et al., 2007) or transfer of the virions from antigen presenting cells to CD4^+^ T cells (Bouhlal et al., 2007). Our conclusion is that the usage of CCR5 remains intact but that the C-HIV and CI-HIV get access to other receptors and induce different signaling pathways that support the higher infection, we have now added this to the manuscript.

4) Figure 4E-F the authors state that infection increases the expression of chemokine receptors, leading to more susceptible cells in the culture, but it would be important to show the chemokine receptor expression in the HIV+ versus HIV- populations for each tissue to clarify whether this is due to a bystander effect or if only those cells that are directly infected increase chemokine receptor expression, which could indicate an alternate interpretation for why the numbers of infected cells also increase.

We agree with the reviewer that it would be interesting to establish the chemokine expression pattern on the HIV positive and negative CD4^+^ T cells. To clarify, we think the HIV exposure and infection of some of the immune cells give rise to an environment that conditions the T cells with more inflammatory factors, which leads to an increased activation/effector status of the T cells, enhanced transcription factors Tbet, and RORgt, and other molecules that regulate the CXCR3, CCR4, and CCR6 chemokine expression. The chemokine receptors can be regulated by cytokines, inflammation, the activation status of the T helper cells, and changes through the processes of cell differentiation, activation, and polarization (Bennett et al. Immunology 2011). For instance, CXCR3 is found on T cells in inflamed tissue and is increased on effector T cells. In addition, T-bet imprints the migratory program upon developing effector T cells and upregulation of CXCR3 (Szabo et al. Science 2002). CCR6 expression on Th17 cells, require the transcription factors RORγt and RORα (Yamazaki et al. J Immunol. 2008). Taken together, this strongly indicates that the activation of and the increase in TH1TH17 and TH17 cell populations by the environment created by the HIV exposure lead to the increased expression of CXCR3, CCR6, and CCR4.

We can rule out the direct effect on chemokine receptor expression caused by productive infection seeing that the increase in expression of CXCR3+CCR6+CD4^+^ T cells goes from 28% on mock to 63% for free HIV, 44% C-HIV and 51% CI-HIV, which is much higher than the levels of HIV-1 p24+ CD4^+^ T cells we can detect. In the case of CCR4+ CCR6+ CD4^+^ T cells, the HIV exposure increased the levels from 16.2% to 33% F-HIV, 30% C-HIV and 34% CI-HIV, which is also higher than the amount of HIV-1 p24+ CD4^+^ T cells that we detected.

We can speculate that the HIV p24+ CD4^+^ T cells seen, should be found among the original colorectal mucosal CXCR3+CCR6+ (TH1TH17) or CCR4+CCR6+ (TH17) CD4^+^T cells. The antibody panel used to evaluate the HIV-1 p24+ cells, i.e. DCs, macrophages and T cells did not include the chemokine receptor antibodies, thus from the existing data we cannot determine if the HIV-1 p24+ CD4^+^ T cells are among the chemokine positive cells positive cells. This is a very relevant experiment that will be feasible to perform with CyTOF/40 color flow that allow more markers to be included in the same panel and will be included in the following study.